# Assessment of Sustainable Ethanolamine-Based Protic Ionic Liquids with Varied Carboxylic Acid Chains as Corrosion Inhibitors for Carbon Steel in Saline Environments

**DOI:** 10.3390/molecules30051033

**Published:** 2025-02-24

**Authors:** Caio Victor Pereira Pascoal, Lucas Renan Rocha Da Silva, Mauro Andres Cerra Florez, Thiago Roberto Felisardo Cavalcante, Julian Arnaldo Avila, Francisco Carlos Carneiro Soares Salomão, Eduardo Bedê Barros, Francisco Avelino, Diego Lomonaco, Regiane Silva Pinheiro, Hosiberto de Sant’Ana, Mohammad Rezayat, Antonio Mateo, Gemma Fargas, Walney Silva Araújo

**Affiliations:** 1Department of Metallurgical and Materials Engineering, Federal University of Ceará (UFC), Fortaleza 60440-900, CE, Brazil; 2Department of Materials Engineering, São Carlos School of Engineering (EESC), University of São Paulo (USP), São Carlos 13566-590, SP, Brazil; thiagorfc@usp.br; 3Department of Aeronautic Engineering, São Paulo State University (UNESP), São João da Boa Vista 13876-750, SP, Brazil; julian.avila@unesp.br; 4Science and Technology Center, Ceará State University (UECE), Fortaleza 60714-903, CE, Brazil; carlos.salomao@uece.br; 5Department of Physics, Federal University of Ceará (UFC), Fortaleza 60455-760, CE, Brazil; 6Department of Chemistry, Federal University of Ceará (UFC), Fortaleza 60440-900, CE, Brazil; 7Department of Food Engineering, Federal University of Maranhão (UFMA), Imperatriz 65915-060, MA, Brazil; 8Department of Chemical Engineering, Federal University of Ceará (UFC), Fortaleza 60440-554, CE, Brazil; hbs@ufc.br; 9Department of Materials Science and Engineering, School of Engineering of Barcelona (EEBE), Universitat Politècnica de Catalunya (UPC), Center for Structural Integrity, Reliability and Micromechanics of Materials (CIEFMA), 08019 Barcelona, Spain; mohammad.rezayat@upc.edu (M.R.); antonio.manuel.mateo@upc.edu (A.M.); gemma.fargas@upc.edu (G.F.); 10Barcelona Research Center in Multiscale Science and Engineering, Universitat Politècnica de Catalunya (UPC), 08019 Barcelona, Spain

**Keywords:** protic ionic liquids, corrosion inhibitor, carbon steel, electrochemistry, saline environment

## Abstract

The inhibitory performance of three distinct protic ionic liquids (PILs), namely, 2-hydroxyethyl ammonium formate (PIL 01), 2-hydroxyethyl ammonium propionate (PIL 02), and 2-hydroxyethyl ammonium pentanoate (PIL 03), was evaluated to determine their suitability as eco-friendly corrosion inhibitors for carbon steel (ASTM A36) in a 3.5 wt. % NaCl aerated neutral electrolyte solution. Standard corrosion inhibitor assessment methods, including electrochemical impedance spectroscopy (EIS), potentiodynamic polarization (PDP), weight loss measurements, and microscopic techniques (SEM and optical microscopy), were employed to examine the steel surface and corrosion rate. There is a general agreement that the inhibition efficacy is directly associated with the adsorption capacity of substances on the surface of an investigated material, normally stainless or carbon steel. The standard free energies of adsorption were approximately −22 kJ mol^−1^, indicating a physical adsorption type of interaction between ionic liquids and the electrode surface. The adsorption behavior of protic ionic liquids on an A36 steel surface conforms to a Langmuir-type isotherm. In conclusion, PIL 01 demonstrated an inhibition efficiency exceeding 80%, while PILs 02 and 03 exhibited efficacies in the 50–60% range. The inhibition efficiency was observed to be proportional to the inhibitor’s concentration. These results suggest that PIL 01, PIL 02, and PIL 03 exhibit significant corrosion inhibition properties.

## 1. Introduction

Corrosion management of metallic materials is essential for the structural stability of industrial processes in general [1,2,3]. The industry resorts to applying low-alloy steel, such as carbon steel (ASTM A36) [4], as an economical alternative to combat damage provoked by degradation occasioned by corrosion [5,6,7]. This specific material is resistant to procedures associated with high-temperature and pressure conditions; however, it has low chemical resistance against corrosion degradation due to its composition [8,9]. For the production of a reasonable substitute, the application of alloys with a low production cost can be evaluated as a plausible choice for the sector; among the various alloy options available in the metallurgical market, carbon steel has gained prominence in the industrial sector due to its favorable mechanical properties and cost benefit.

Thus, faced with numerous alternatives against corrosion, the industry has recently opted to apply corrosion inhibitors, which are chemical compounds from natural or synthetic sources that can prevent or eliminate corrosive processes [7,10,11,12,13]. The effectiveness of a corrosion inhibitor depends on the ability of this compound to fill voids on a material’s surface, which are the results of the atomic interaction between the studied compounds and the metal. Generally, this primary interaction forms a protective film with the metal via chemisorption or physisorption, protecting the surface against corrosion damage [14,15,16].

Due to the significant demand for corrosion inhibitors that are technically, economically, and environmentally viable for industrial applications, ionic liquids (both aprotic and protic) offer a promising alternative. Aprotic ionic liquids (AILs) are particularly versatile because of their high ionic character; modifying their synthesis by changing either the anion or cation can yield new products with distinct properties.

In general, ionic liquids are less harmful to the environment than traditional corrosion inhibitors containing heavy metals in their composition. In addition to this factor, no toxic solvents are used in their production [6]. However, the high thermal stability of AILs has a distinct and crucial disadvantage, as they bioaccumulate in animal species and soil [17,18,19,20,21,22,23,24,25]. Therefore, due to the complexities associated with AILs, it is necessary to explore potential corrosion inhibitors. Hence, protic ionic liquids (PILs) were investigated as a new and plausible material, as they can exhibit all the promising properties. However, they benefit from being more accessible to synthesize and cheaper to produce than AILs [26], particularly since they are biodegradable [27].

Protic ionic liquids (PILs) represent a significant advancement in corrosion inhibition, offering distinct advantages due to their unique properties and environmentally friendly nature. Unlike conventional inhibitors, which often rely on hazardous metals and toxic solvents, PILs are synthesized through straightforward and cost-effective processes without harmful solvents, making them an attractive and sustainable alternative [17,18,19,20,21,22,23,24,25]. Moreover, their highly adaptable chemical structure allows for precise modifications to enhance the desired properties. This versatility, combined with their biodegradability and low environmental impact, makes PILs a promising solution to meet the growing industrial demand for efficient, sustainable, and multifunctional corrosion inhibitors [27].

Despite extensive research on metallic corrosion in metallurgy and the development of protection strategies, studies focused on carbon steel, particularly the carbon steel alloy (ASTM A36), remain scarce in the literature on metals [28,29]. ASTM A36 steel is extensively utilized in industry due to its structural properties, excellent mechanical strength, and broad market availability. While its corrosion behavior has been widely investigated, its interaction with specific sustainable inhibitors, such as protic ionic liquids, in aggressive environments like saline conditions remains insufficiently explored in the literature. Therefore, addressing this gap is crucial in metallurgy studies, especially with regard to establishing the interaction between severe environmental conditions and the formation of corrosion products, influencing the performance and longevity of metallic materials.

Applying protic ionic liquids (PILs) as corrosion inhibitors for ASTM A36 carbon steel in saline environments represents an innovative and promising approach that addresses significant gaps in the extant corrosion literature. Indeed, by focusing on this material, which is extensively employed in structural and industrial contexts, this study addresses the paucity of research on its corrosion behavior under aggressive conditions. PILs introduce a novel protective mechanism that enhances durability while aligning with principles of sustainable material science and metallurgical engineering. This research contributes to the understanding of corrosion mechanisms in ASTM A36 but also elucidates the potential of this option, the PILs, as a transformative solution for mitigating material degradation in diverse industrial applications.

From this discussion, investigating ASTM A36 aligns with sustainable practices by prioritizing cost-effective and abundantly available materials in terms of market. Therefore, its wide application in the construction, transportation, and energy industries highlights the importance of increasing corrosion resistance through innovative strategies. This contributes to reducing resource consumption and minimizing the environmental and economic costs associated with material degradation and replacement. The use of sustainable corrosion inhibitors is necessary to meet the demand for sustainable and durable solutions in engineering and industrial contexts. This research investigates the potential of protic ionic liquids for controlling corrosion, employing electrochemical techniques on ASTM A36 carbon steel alloys under saline conditions and replicating the operational environments found in manufacturing processes.

## 2. Results and Discussion

### 2.1. Anticorrosive Performance of PILs

#### 2.1.1. Weight Loss Measurements (WL)

The corrosion rate (νcorr) and the inhibitor efficiency (% IE) of the carbon steel sample (A36) in the saline electrolyte were determined by Equations (9), (10), and (11), respectively, (Section 3.4.1) and detailed in Table 1. In the mass loss experiments conducted in this study, two distinct scenarios were analyzed. The first one, in which PIL 01 remained promising up to the highest concentration of 1000 ppm, achieved efficiency values exceeding 82%. However, a distinct situation was observed for PIL 02 and 03, where the best inhibition efficiency was reached with the middle concentration parameter of 500 ppm, and, interestingly, with the increase in concentration, a significant reduction in efficiency was observed.

#### 2.1.2. Adsorption Isotherm and Thermodynamic Analysis

The protective efficacy of corrosion inhibitors was influenced by their capacity for adsorption on the surface of metallic materials. Some aspects, such as the nature of the aggressive ambiance, inhibitor structures, species charge distribution and metal charge, determine the adsorption process in most cases. Typically, corrosion inhibitors adsorb on the surface by physical or chemical adsorption processes, which reduce the reaction area susceptible to corrosive attack and protect the material for a specific period of time [30,31]. 

Hence, to appraise the adsorption process and interaction between the protic ionic liquids and the steel surface, the common isotherms were verified using the Langmuir, Freundlich, Temkin, and Frumkin equations, as follows [11]:(1)$$\frac{\theta }{\left(1-\theta \right)}=K_{ads.}\;\cdot \;{CPILs}_{conc.}$$(2)$$Log\;\theta =Log\;K_{ads.}+n\;Log\;{CPILs}_{conc.}$$(3)$$\theta =\left(\;\frac{1}{f}\;\right)\ln K\;\cdot \;CPILconc.$$(4)$$\left[\frac{\theta }{1-\theta }\right]e^{f\theta }=K_{ads.}\;{\cdot \;CPILs}_{conc.}$$

Here, *θ* = surface coverage; *CPIL_conc_*_._ = concentration of PILs (ppm); *K_ads_*_._ = adsorption equilibrium constant (mol^−1^); *f* = *PILs* interaction parameter; *n* = number of H_2_O molecules replaced by *PILs* molecules; and *K* = constant [11]. 

Then, the three isotherms tested fit well with the experimental data. For the Langmuir isotherm (Figure 1), the best correlation coefficient (R_2_) value obtained was PIL 01 = 0.9985, PIL 02 = 0.9997, and PIL 03 = 0.9853. Meanwhile, for the Frumkin isotherm, PIL 01 = 0.9311, PIL 02 = 0.9911, and PIL 03 = 0.4622. Also, for the Temkin isotherm, PIL 01 = 0.9062, PIL 02 = 0.9975, and PIL 03 = 0.4836, respectively.

Therefore, given the visible difference in the values of the linear coefficients among the models evaluated (Langmuir, Frumkin, and Temkin), the Langmuir model was assumed to exhibit the best behavior for the electrochemical system.

The results for both isotherms indicate the existence of an attractive lateral interaction in the adsorption layer. The inspection of interactions between the investigated inhibitor and the metal surface is crucial, as these secondary results can justify side interactions, such as those explained by adsorption isotherms [32,33,34]. The free energy of adsorption (ΔG also) was calculated from the Langmuir isotherm since it showed the best correlation with the experimental data, according to Equation (5), as follows [12]:(5)$${∆G}_{ads.}^{0}=-R\cdot T\cdot ln(55.5\;Kads)[kJ/mol]\;$$

Here, T is the absolute temperature, R is the universal gas constant, and 55.5 represents the water concentration in moles [35,36]. The adsorption–desorption equilibrium constant K was determined as 0.0103 for PIL 01, 0.0107 for PIL 02, and 0.0042 for PIL 03, which is a unit of measurement (L·mg^−1^), leading to ${∆G}_{ads.}^{0}$ = −22.89, −22.98, and −20.67 by kJ mol^−1^. It is a standard in the literature [37,38,39,40] that negative values of ${∆G}_{ads.}^{0}$ denote adsorption by a spontaneous process. It is crucial to evaluate data such as ${∆G}_{ads.}^{0}$ because this analysis assists in establishing the type of adsorption between the metal surface, specifically A36, and the inhibitor. It is generally accepted that ${∆G}_{ads.}^{0}$ values close to −20 kJ/mol or higher have a lower adsorption energy of lower intensity.

This indicates that the electrostatic interaction between the molecules in an electrolytic NaCl solution and the metallic surface, known as adsorption by physisorption, is probably explained by the ionic nature of the compound [39,40]. Finally, the values of K_ads_ are reasonably low, providing significant information about the chemical interaction of the adsorbing molecules of protic ionic liquids with (A36); the interaction with the carbon steel surface is represented by physical adsorption.

According to the literature on corrosion inhibitors, similar physical adsorption results have been reported, with ${∆G}_{ads.}^{0}$ values around −20.0 kJ·mol^−1^ and ranging from −16.0 to −19.0 kJ·mol^−1^ [41,42]. This reinforces that, although less frequently highlighted, inhibitors relying on physisorption are well-documented in corrosion protection studies [41,42].

#### 2.1.3. Open Circuit Potential Measurements (OCP)

Evaluating the potential value versus time constitutes a methodology for monitoring interface changes between a metal and the electrolyte solution [43]. Initially, the system was placed with agitation for 1800 s [37,44], and after this period, this agitation was turned off and the potential continued for another 1800 s [45,46,47].

Interestingly, it was observed in the system that, during constant agitation, there is a slight reaction with a reduction in the electrochemical potential due to the turbulence between the ionic compounds and the metallic surface. After a period, the system stops stirring, causing a sudden change in the reaction followed by potential stabilization (Figure 2). The stationary open circuit potential (E_ocp_) was attained by the curves representing the interaction between the sample and the sodium chloride electrolyte with PILs at different concentrations (Figure 2).

This representation exemplifies the clear disparity between the values of E_ocp_ following the addition of PILs with more positive values compared with the blank, (−0.384 PIL 01–1000 ppm), (−0.425 PIL 02–500 ppm), (−0.452 PIL 03–250 ppm), and (−0.498 Blank) as a function of time. The system was applied in two distinct conditions to confirm the capacity of the potential reduction by the ionic compounds.

According to Zeino et al. [43], in experiments that display a disparity between the inhibitors evaluated and the blank, this disparity is attributed to the high electrochemical activity of the carbon steel in aerated 3.5% NaCl media. This information is also observed in the OCP evaluation with PIL experiments, as Figure 2 confirms this difference [48].

#### 2.1.4. Potentiodynamic Polarization Measurement (PDP)

The polarization curves were determined from different concentrations (250, 500, and 1000 ppm) of the three protic ionic liquids on the potentiodynamic behavior of ASTM A36 carbon steel in a 3.5% wt. NaCl solution at room temperature. Indeed, the influence of the protic ionic liquids in the electrolyte solution is also noticeable in the mechanism of the corrosion reaction, which is represented in the potentiodynamic polarization curves in Figure 3. This addition is particularly noteworthy as it affects both the cathodic and anodic branches of the polarization curve. Also, the observed results qualitatively suggest a huge difference in the ability to inhibit.

Still, the polarization curves generally provide a numeric result; the corrosion potential (termed as overpotential), a fixed point that enables the analysis of all branches that extend over the total initial cathodic stages (−0.2 V) to the anodic region (0.2 V), sweeping the potential in a constant rate of 1 mV/s, as shown in Figure 3, where all inhibitors (PIL 01, 02 and 03), was used. This, in turn, indicates that the anodic branch often has smaller current density values (A/cm^2^) compared to the tested inhibitor-free one (blank). 

More specifically, to investigate the polarization technique and the effect of the corrosion inhibitors, the corrosion current densities (I_corr_) and corrosion potentials (E_corr_) were determined by extrapolating the linear regions of the anodic and cathodic Tafel curves, as shown in Table 2. This finding demonstrates the behavior of an anodic-type inhibitor.

Across the entire 250 to 1000 ppm concentration range, the current values for all tested compounds (PILs) were persistently below the blank value, indicating the anodic mode of protection. In addition, the E_corr_ values of the ASTM A36 working electrode in the presence of PILs compared to the blank samples also showed a shift of about 90 mV, determined using a saturated Ag/AgCl reference electrode in KCl. 

The averaged shift indicates that the corrosion potential is displaced more to the positive range (anodic direction), attributed mainly to the polarization of the anodic reaction after the E_corr_ point. Finally, such transition is linked to surface changes in the substrate (ASTM A36 carbon steel) following inhibitor adsorption.

Importantly, in electrochemistry, a compound or glomerate of molecules could be classified as an anodic, cathodic, or mixed corrosion inhibitor by the context and the technique used. By performing potentiodynamic polarization assisted by the Open Circuit Potential (OCP) measurements together, it is possible acquire information about the inhibitors. When the displacement in OCP values exceeds 85 mV relative to the blank (without inhibitor), it indicates a significant change in the potential, allowing for a preliminary categorization of the inhibitor type.

In the study of El-Tabei et al. [49], with surfactants as inhibitors acid electrolytes, the shift value fell within ±85 mV in both branches; this result characterizes a possibly mixed inhibitor type. Also, in the research of Verma et al. [50], using new ionic liquids as inhibitors in acid electrolyte, the limit shift value of E_corr_ was 75 mV, which is less than 85 mV, as in the present study, and is the base value for this discussion.

The electrochemical results obtained from the potentiodynamic study of protic ionic liquids (PILs) are comparable to those reported in the literature for corrosion data regarding the Aprotic type. Additionally, they exhibited lower current densities with the addition of these ionic compounds, with the results being clearly visible in the polarization curves.

Additionally, these results confirm the assessment of these compounds as potential electrochemical corrosion inhibitors in a saline solution, indicating a simulated marine approach (3.5%). The application of these PILs into the system alters the polarization curves of the tested compounds (PIL 01, 02, and 03) (Table 2). This change allows the polarization curves to exhibit a decrease in current densities which facilitates the protection of the material applied in the system [30,51,52].

To improve our understanding of the theoretical principles underlying the interpretation of potentiodynamic polarization tests with the addition of corrosion inhibitors, for the anode inhibitors, the cathodic branch is positioned near the blank curve, while the anodic branch is situated below it, demonstrating a difference in current density (I) when the potential is plotted relative to the electrode area. 

Crucially, this decrease in current density possibly results from the formation of a protective inhibitor layer on the surface, which primarily degrades over time in the anodic region and with variations in the current density within the system [47,53,54]. Due to the occurrence of this type of protection, it is evident from the results of the morphological characterization that the metal surface covered with the inhibitor presents a marked reduction in degradation when compared to the sample without the addition of the inhibitor (PILs).

In Table 2, the elements evaluated are E_corr,_ corrosion potential, and I_corr_, current density, to obtain the efficiency values (Eff.) through the use of inhibitors. The efficiency values presented in Table 2 were determined based on the relationship between the current density values (I_corr_), following a methodology analogous to that reported in the literature on corrosion inhibitors [8,55]. The formation of a barrier layer on the mild steel surface by the inhibitor molecules was favored by the increase in concentration, resulting in a decrease in the electrochemical reaction rates.

#### 2.1.5. Electrochemical Impedance Spectroscopy Measurements (EIS)

The corrosion of carbon steel (A36) in saline solution was studied, both with and without PILs at different concentrations. One of the most reliable and replicated techniques in the electrochemical field is Impedance Spectroscopy (EIS), as it evaluates the efficacy of the protective film over the steel. The EIS results are shown in Figure 4, and the impedance parameters are provided in Table 3.

The EIS graphics are presented in Figure 4, separated into Sections A–C, with the impedance parameters detailed in Table 3. At the initiation, the Nyquist plots reveal a complete capacitive semicircle, demonstrating the interaction between the PIL molecules solubilized in the saline electrolyte and the carbon steel surface. This result suggests a considerable enhancement in the polarization resistance (Rp) induced by the addition of the three protic ionic liquids in the solution. Subsequently, this assumption was derived from the considerable increase in capacitive arcs, contrasting with the blank without the PILs.

Furthermore, the results presented in Figure 3A correspond to the data obtained from the polarization assessment, where PIL 01 (Figure 3A) exhibits higher inhibition efficiency than the other two PILs, 02 and 03, under investigation. Additionally, these results demonstrate the inhibition efficiency of protic ionic liquids (PIL 02 and PIL 03) assessed at different concentrations.

In the circuit (Figure 5), the reference electrode is denoted as R.E., the working electrode as W.E., the solution resistance as Rs, the polarization resistance as Rp, and the constant phase element of the double layer as CPE. The constant phase element, characterized by Y_0_ and n, which correspond to the admittance and the exponent of the CPE, respectively, was included to address the frequency decline phenomenon. Table 3 provides a detailed representation of the fitting results, including the crucial data obtained from the impedance tests.

All concentrations of the inhibitors studied (PIL 01–03) were verified employing distinct circuits. However, a reliable simulation is provided by the Nyquist plots with the equivalent circuit (Figure 5). Specifically, to calculate inhibition efficiency (I.E) and double-layer capacitance (C_dl_), Equations (6) and (7) were employed, respectively, where R_p_ and R_p(inh)_ represent the polarization resistances in the absence and presence of inhibitors [56,57].

In Table 3, the final Rp value of PIL 01 remained prominent, exceeding 1000 Ω·cm^2^ across variations in inhibitor concentration, indicating effective protection against corrosion. However, for PILs 02 and 03, an observed reduction in the capacitive loop occurred as the concentration increased from 500 to 1000 ppm. This reduction is due to the saturation of the electrochemical layer at maximum concentration, behaving similarly to what has been reported in the literature [58,59].(6)$$I.E\left(\%\right)=\frac{\left(Rp-{Rp}_{\left(inh\right)}\right)}{\left(Rp\right)}\times 100$$(7)$$C_{dl}({F\cdot cm}^{-2})={\left(\;Q\times \;R_{P}^{\;\;\;\;1-n}\right)}^{1/n}$$

Moreover, the impedance results demonstrate a noticeable increase in polarization resistance (R_p_) due to the addition of the PILs, as demonstrated by the larger capacitive arcs compared to the blank sample without the corrosion inhibitors. In addition, the results in Figure 4 contrast with the observations from potentiodynamic polarization (Figure 3 and Table 2), indicating that PIL 01 exhibits a more pronounced inhibitory effect than PILs 02 and 03. The detailed examination and discussion of the recorded, obtained, and calculated EIS data presented in Table 3 reveals that the Rs values for ASTM A36 increase with the addition of PIL molecules in saline solution. This behavior is characterized by a type of chemical compound that can be used as a corrosion inhibitor, representing the ability to protect the material when added to the electrolytic medium. 

The solution resistance (Rs) is observed at concentrations as low as 250 ppm (3.90 Ω·cm^2^), compared to the blank sample (1.78 Ω·cm^2^), and it continues to rise, reaching a maximum value of 4.29 Ω·cm^2^ at 1000 ppm.

These results are reflected in the adsorption process of PILs on the surface. They formed a stable and protective layer on the surface of the ASTM A36 acting as a barrier against the aggressive attack of the chloride ion (Cl^−^), which is mainly present in the NaCl electrolyte simulating seawater, thus reducing the corrosion rate of the steel with its addition in the solution. The EIS results indicate that increasing Rs decreases C_dl_, attributed to the substitution of water and aggressive ions (high ε) by the adsorbed PIL molecules (low ε). Moreover, an increase in the thickness of the protective film (T) over the steel surface was observed, according to the Helmholtz model.

In a thorough comparison of the data obtained from the techniques employed in this study, including weight loss, potentiodynamic polarization, and electrochemical impedance spectroscopy, it becomes clear that, under the experimental conditions involving 3.5 wt. % NaCl, an initial concentration of 250 ppm remains insufficient to protect the material, as corrosion values exceed 40%. While these preliminary values are relatively low, they suggest a promising trend for the use of these corrosion inhibitor compounds due to their protective properties. Additionally, it is essential to emphasize the protective capability of PIL 01, which, even at the minimum concentration, achieves efficiency results exceeding 60%, contrasting with the other PILs.

Moreover, at the intermediary concentration (500 ppm), the efficiency of the three evaluated inhibitors increased significantly, with PIL 01 exhibiting a solid performance. This increase highlights one of the main factors analyzed, the size of the carbon chain, which particularly affects the possibility of interatomic interactions on the steel surface with the addition of PILs.

At the highest concentration investigated, 1000 ppm, PIL 01 again demonstrated superior inhibition efficiency, reaching over 80% in a neutral/saline solution. However, a distinct tendency emerged with PIL 03, which, unlike the other inhibitors, displayed a marked decrease in inhibitory effectiveness at this higher concentration. This reduction could be attributed to the saturation of the metallic surface by the larger carbon chain of PIL 03, which led to a reversed effect, decreasing substantial protection instead of enhancing it. This finding indicates that, although increasing concentration generally improves efficiency, certain molecular characteristics can produce unforeseen results.

### 2.2. Critical Micellar Concentration (CMC)

The critical micelle concentration (CMC) is an important/distinct point at which the concentration of an assumed surface-active agent in solution alters the initial state of molecular solvation. At this concentration, several of the chemical system’s physical and chemical properties undergo significant changes. This phenomenon is crucial for theoretical studies and has practical applications, such as enhancing the understanding of corrosion inhibitor systems [60,61].

Figure 6 illustrates two distinct scenarios, where PIL 01 has an extensive later stabilizing concentration period (after 620 ppm) compared to PILs 02 and 03. This factor is related to a significant difference in the size of the carbon chain. Since formic acid is the smallest carboxylic acid utilized in the ionic liquid structures investigated in this study, a considerable amount of this compound combination is necessary to saturate the interaction in the metal. 

The moderately higher value of CMC of PIL 02 may explain this, because the finest concentration of this inhibitor was obtained at 1000 ppm, considering that above the middle value of the concentration around 600 ppm, the electrolytic system was already saturated by the miscella formation. 

The remaining protic ionic liquids (02 and 03) demonstrated a reduction in corrosion control efficiency at the highest concentration tested, possibly due to system saturation at levels below 400 ppm (340 ppm for PIL 03 and 360 ppm for PIL 02). This type of result is essential for understanding the reaction mechanism when there is an adsorption of inhibitor molecules on the surface of the steel studied. 

A critical aspect of this discussion is the carbon chain length of protic ionic liquids (PILs), as it significantly influences their inhibition efficiency. Molecular size directly affects interactions with the metal surface, adsorption mechanisms, and overall protective performance, making it a key parameter in evaluating the effectiveness of these compounds.

### 2.3. Surface Analysis

In this section, the importance of surface assessment was discussed. Typically, the morphology of metallic materials, such as carbon and stainless steel, is examined as a visual and partial method to gauge the inhibition efficiency of these compounds, protic ionic liquids (PILs), both before and after the electrochemical tests [62,63]. 

For instance, evaluating morphology through an optical microscope provides essential insights into the material, providing a preliminary overview. This information is essential for assessing performance in experiments, including indicators such as excessive oxide/oxyhydroxide formation, modifications in solution color, and the solubility of the inhibitor in the electrolyte.

#### 2.3.1. Morphology Evaluation of Steel by Optical Microscopy (OM)

Figure 7 presents micrographs of the ASTM A36 specimens exposed to a 3.5 wt. % NaCl solution, with and without corrosion inhibitors, after three distinct periods (24, 48, and 72 h). In sum, after the first period of evaluation (24 h), material corrosion became more pronounced, as indicated by the roughened surface in the first row and the presence of corrosion products, oxides, and oxyhydroxides on the surface. As a result, this high level of deterioration can be ascribed to carbon steel’s heightened vulnerability to corrosion in saline environments, and this can be attributed to the low chromium content of the alloy [64]. Consequently, implementing anti-corrosion strategies, such as sustainable corrosion inhibitors, is essential for protecting the material’s surface.

Moreover, after the initial immersion period of evaluation, the general corrosion of the carbon steel intensifies more rapidly, as demonstrated by the rough surface observed in the first row. This emphasizes the real vulnerability of carbon steel in a saline environment potentialized by the aggressivity of the chloride ion [46]. Macroscopically, certain areas expose a noticeable absence of significant corrosion products, indicating the effective action of protic ionic liquid on the surface to form a protective film [65]. These findings were further corroborated by the scanning electron microscopy tests in the subsequent section of the material surface analysis.

Additionally, during the evaluation of the three protic ionic liquids, certain areas remain protected within the initial 24-h period of immersion. However, when the assessment period is extended to the mildest period of evaluation (48 h), the corrosion process intensifies drastically, resulting in a substantial increase in oxide formation on the material, except for PIL 01, where there is still some partial protection of areas distributed on the surface (Figure 7). 

This specific compound, synthesized with formic acid and ethanolamine, continues to exhibit clean areas free from corrosion products, confirming its effectiveness as the most promising compound in this study. The literature emphasizes the critical role of corrosion assessments with corrosion inhibitors in shaping the future of the industry. In this context, recent studies provide a deeper understanding of the interactions between chemical compounds—whether aprotic or protic—and metal surfaces, contributing to more comprehensive insights into corrosion mechanisms. Correspondingly, Guo, H. et al. [66] utilized the same evaluation to investigate the interaction of protic ionic liquids as lubricants. This external perspective of the material is crucial for determining how the active composites of the material can interact, facilitating their evaluation in specific tests.

#### 2.3.2. Scanning Electron Microscopy Analysis (SEM)

Scanning electron microscopy (SEM) was applied to examine the material’s morphology following polarization tests conducted with 1000 ppm, and the best concentration for each protic ionic liquid was evaluated. [65]. ASTM A36 steel specimens were immersed in a 3.5% NaCl electrolyte solution under two distinct conditions as follows: (A) without inhibitors and (B) with the addition of PIL 01 (HEAF), (C) PIL 02 (HEAP), and (D) PIL 03 (HEAPe). These conditions were maintained during a one-hour Open Circuit Potential (OCP) and potentiodynamic polarization test at a room temperature of 25 °C.

Carbon steel in a 3.5% NaCl solution without an inhibitor (Figure 8A) exhibits an irregular and deteriorated surface, indicating the severe degradation of the outer layer of the evaluated material due to the saline and aggressive environment, as noted in the literature on corrosion inhibitors [42,47,65,67,68,69,70]. In contrast, it is observed that a significant reduction in surface damage was observed, with a more uniform appearance and less evidence of corrosive damage. These changes were observed in the presence of all evaluated PILs (Figure 8B–D).

Additionally, only a minimal amount of corrosion products has formed on the sample, with certain areas revealing no signs of corrosion. A fragile protective film is noticeably adsorbed on the surface [67]. These results provide strong evidence for the effectiveness of Protic Ionic Liquids (PILs) on ASTM A36 carbon steel in a saline environment. Furthermore, the steel sample immersed in a 3.5% NaCl solution containing PIL 01 (Figure 8B) exhibits a smoother surface. This result is consistent with the inhibition performance observed in the weight loss and electrochemical tests.

#### 2.3.3. Atomic Force Microscopy (AFM)

Atomic Force Microscopy (AFM) is an effective and reliable method for analyzing surface morphology and rugosity to understand the corrosion inhibition process. Specifically, three-dimensional (3D) micrographs were obtained to analyze ASTM A36 steel samples exposed to both inhibited and uninhibited corrosion conditions, using the inhibitor with the best performance, PIL 01. When the inhibitor was absent, the steel surface exhibited roughness with the formation of deep plaques.

The absence of the selected inhibitor in the electrolytic solution caused significant surface degradation within 24 h of immersion, due to the aggressive nature of the environment, which was intensified by chloride ions. Moreover, prior electrochemical and gravimetric tests, as described in Section 3.1, confirmed that PIL 01 demonstrated the most effective corrosion protection. These findings underscore the efficiency of PIL 01 in corrosion inhibition and highlight its potential as a promising candidate for corrosion protection in practical applications.

According to these findings, it is possible to approximate a detailed average roughness profile of the surface. This information is crucial for the study of corrosion inhibitors, as it enables the assessment of surface damage on a nanometric scale following immersion in an electrolytic solution (3.5% wt. NaCl at 25 °C), both with and without the presence of PILs. The AFM results confirm and establish a correlation suggesting that the incorporation of chemical compounds, acting as corrosion inhibitors, reduces surface roughness. Figure 9 offers an alternative approach for evaluating the rough profile of metal surfaces. Phase images, in particular, provide a comprehensive view of the material using Atomic Force Microscopy (AFM). In this analysis, specific regions of the micrograph were highlighted using a fixed height limit of 150 nm, facilitated by Gwyddion V. 2.6 software.

This information is essential for evaluating the true condition of the material after exposure to the electrolytic medium (3.5 wt. % NaCl). In the graph for the blank sample (without inhibitor), areas with elevations above 150 nm are evident, reflecting significant wear on the surface. Furthermore, the control sample (polished) and the most efficient inhibitor, PIL-A, show minimal surface areas exceeding 150 nm, with the control surface (polished) exhibiting no such elevations. A comparison of these results with the 3D images highlights the protective effect of the material when exposed to the electrolytic medium with the addition of the protic ionic liquid. In contrast, the average surface roughness (A.S.R.) for the uninhibited samples (NaCl 3.5%/24 h) was 185 nm (Figure 9C). However, for PIL 01 at its optimal concentration (1000 ppm), the roughness was significantly reduced to 11.8 nm (Figure 9A), closely resembling the control surface value (8 nm) used for the baseline evaluation (Figure 9B).

As shown in Figure 9, there is an apparent increase in both the roughness and thickness of the carbon steel surface after 24 h in saline electrolyte. This information indicates that there is a clear and thin protective layer on the metal surface, likely attributed to a physisorption process attributed to the value of free Gibbs energy. Moreover, the chemical structure of PILs appears to be capable of absorbing onto the surface by Van der Walls bonds defined by weak intermolecular forces arising from electrostatic interactions between adjacent molecules, partially blocking the harmful effects of Cl^−^ ions.

Therefore, these results support the conclusions presented earlier in Section 2.1.2. This protective film effectively shields portions of the material’s surface, an effect facilitated by the application of the corrosion inhibitor (PIL 01) at a maximum concentration of 1000 ppm.

PIL 01 exhibited a protective effect against corrosion when adsorbed onto the material’s surface in a 3.5 wt. % NaCl solution. Figure 10 presents the 2D roughness profile from AFM images, highlighting a distinct contrast between carbon steel samples exposed to the saline medium without the inhibitor and those treated with 1000 ppm of PIL 01.

The roughness profile reveals notable differences in surface texture, emphasizing the inhibitor’s influence on mitigating surface deterioration. The presence of PIL 01 appears to reduce corrosion-induced roughness, suggesting its effectiveness in modifying the steel’s topography under aggressive conditions. These findings underscore the inhibitor’s role in preserving surface integrity and minimizing corrosion-related degradation in saline environments.

#### 2.3.4. Characterization of Corrosion Products by X-Ray Diffraction (XRD)

X-ray diffraction (XRD) is a widely used technique for analyzing the composition of corrosion products, providing essential insights into the formation and evolution of oxide/oxyhydroxide layers on metal surfaces. Understanding these layers is critical for assessing the effectiveness of corrosion inhibitors. In this study, XRD was employed to evaluate the influence of protic ionic liquids (PIL 01–03) on corrosion mitigation.

The inhibitors were tested at a concentration of 1000 ppm over 30 days of immersion, allowing for a detailed assessment of their impact on the structural characteristics of the corrosion products. This approach enables a comprehensive evaluation of the inhibitors’ performance in altering oxide layer formation and improving corrosion resistance in aggressive environments.

Initially, the XRD analysis revealed the existence of oxides and oxyhydroxides, along with a surplus of NaCl (halite). Thus, as already described in the literature, due to the saline nature of the electrolyte used, the presence of this specific phase is acceptable because the concentration is very high and the long period of immersion (30 days), approximately 35.000 ppm, is equivalent to 3.5% by weight of NaCl. In addition, the results offer valuable information about the material deposition process, considering the formation of oxides and the deposition of crystals from the solution.

In this context, there is a significant difference in the phases found in the diffraction patterns of the blank sample (without inhibitors) compared to the protic-ionic liquids (inhibitors) (Figure 11). The main variance observed in the diffraction study with the addition of the corrosion inhibitors (PILs) is the presence of a peculiar phase of corrosion products that serve as an indication of oxyhydroxide formation, specifically Goethite (Table 4).

In corrosion studies, especially in aqueous media, two main conditions are typically considered as follows: aerated and non-aerated aqueous solutions. This research conducted experiments within an open system, for example, using electrochemical and mass loss tests, to simulate an aerated situation, aiming to elucidate the mechanism of corrosion. The X-ray diffraction technique is a fundamental analytical method for assessing oxides and hydroxides in systems where carbon steel is exposed to elevated levels of oxygen and chloride.

Furthermore, lepidocrocite, magnetite, goethite, and halite are the primary corrosion product morphologies observed by the interpretation of the diffractogram. Additionally, magnetite, lepidocrocite, and halite were present in all samples, including those with and without corrosion inhibitors, indicating the universal nature of these compounds in the studied system. The analysis of corrosion products offers essential insights into the environment through the formation of specific oxide classes. This approach provides the academic community with valuable information on the behavior of the electrolytic system under aerated conditions with the application of protic ionic liquids as corrosion inhibitors.

In addition, detecting the formation of distinct phases when applying an inhibitor over extended periods can elucidate the interactions between the corrosion inhibitors and the steel surface. Through this analysis, it becomes possible to deepen our understanding of how inhibitors influence the development and stability of corrosion products, specifically goethite, thereby shedding light on the effectiveness of these inhibitors in protecting steel surfaces in corrosive ambiance [71].

A goethite-based internal layer forms on the metallic surface, characterized by an amorphous structure that is stable, compact, and dense. Due to these structural attributes, this phase provides a barrier/protective quality to the metal surface, as indicated by studies in the literature on atmospheric corrosion [72]. Independently, atmospheric corrosion research often spans lengthy periods; however, a critical consideration is that the presence of specific phases can enhance the material’s protective ability due to the compact and dense behavior of this phase. This blockade can appear with the addition of an inhibitor or the application of an anti-corrosion coating, both of which contribute to the material’s corrosion resistance over time [73].

PIL 01 exhibited the highest corrosion resistance, as confirmed using electrochemistry, weight loss, and microscopy techniques. Interestingly, the phases identified in the blank sample also appeared in the spectrum of PIL 01 (Figure 11), although with a notable difference as follows: the presence of the goethite phase in the spectrum of PIL 01. This indicates that the enhanced protection of PIL 01 is likely attributed to the formation of a denser, more compact layer on the surface of the A36 Carbon Steel. After a 30-day immersion, this protective layer achieved efficiency values above 85%, demonstrating effective material protection and confirming the potential of the inhibitor for extended applications in corrosive environments.

Typically, various oxides and hydroxides can be found in corrosion layers, such as Lepidocrocite, Magnetite, Hematite, and Goethite. Nonetheless, the formation of these specific phases can be related to external factors, such as pH and temperature, in saline immersion; in this investigation, only lepidocrocite and magnetite were commonly located above the goethite layer. The formation of lepidocrocite and magnetite on the outer layer of goethite over a metallic substrate is generally linked to saline atmospheric conditions [72]. This combination of phases exhibits unfavorable properties due to the high proportion of the material which allows contaminants such as O_2_, Cl^−^, and Na^+^ from the environment to permeate. This permeation accelerates the corrosion process, resulting in prolonged and more rapid material degradation over time.

In a discussion about layers in saline electrolyte systems where denser internal oxide layers such as goethite are formed, some studies propose that this layer can exert a protective effect on steel, including both carbon and stainless steel. The goethite layer may act as a hardy barrier, effectively reducing the permeation of contaminants and helping to maintain the material’s structural integrity. Significantly, this oxide phase was observed in the XRD spectrum of the most efficient corrosion inhibitor tested in this study (PIL-A), underscoring its potential role in corrosion resistance [74].

Regarding the phases identified in the XRD analysis, in literature it is possible to associate magnetite with the normal formation of a corrosion layer on the surface, offering the least protection on steel surfaces. Even so, numerous researchers have emphasized the protective properties of goethite and ferrihydrite in relation to metal surfaces; however, in this article, experiments involving Protic Ionic Liquids (PILs) revealed only the presence of goethite, and the discussion on the influence of the addition of corrosion inhibitors in solution on the formation of these oxides is quite scarce in the literature on inhibitors and corrosion, thus emphasizing the importance of this investigation) (Figure 11) [75].

Furthermore, hematite and magnetite exhibit limited protective effectiveness capacity over the surface, attributable to their high porosity and instability. Furthermore, high porosity is a crucial factor in the study of oxides, since the coating with permeable characteristics facilitates the entry of ions with access to the metallic matrix, a negative highlight for the acceleration of the base metal to the chloride ion. In sum, this characteristic reduces their suitability for corrosion resistance, as their structural properties hinder the formation of a stable, protective layer.

Two distinct scenarios were identified for PILs 02 and 03. Both samples of protic ionic liquids presented the same oxide/oxyhydroxide phases found in the control sample (without PILs). As summarized in morphological studies, the formation of specific iron oxide/oxyhydroxide phases is directly influenced by the pH and temperature of the electrolytic environment [76]. Therefore, depending on the conditions encountered, the rust found above the surface can evolve to form stable phases such as goethite. For example, ferrihydrite, considered a transient phase in aqueous conditions, gradually endures transformation into a stable, solid, and thermodynamically favorable crystalline phase. This phase can develop into either goethite or hematite, depending on the reaction medium’s pH and temperature, aligning with the literature [77].

Still, the modifications of the corrosion products formed on the metal surface typically occur under controlled conditions, such as specific temperature ranges (27 to 70 °C) and pH values (7.5 to 9.0), because the modification of oxides at alkaline pH and high temperatures occurs due to the higher concentration of OH^−^ ions and the increase in thermal energy, which accelerates oxidation. These conditions favor the formation of stable oxides, such as goethite and hematite, and alter the structure and morphology of the oxides, influencing their corrosion properties and adhesion to the metal surface [78].

There is a reasonable basis for the development of distinct phases on the metal surface, particularly goethite and ferrihydrite [79,80]. During the morphological assessment of the steel surface, pH and conductivity were measured after the addition of all PILs to the NaCl solution (3.5% wt.) over periods of 24, 48, and 72 h. Results indicated that the pH of the solution containing PIL 01 initially decreased from 8.4 to 6.7 over 72 h. Within this timeframe, the pH remained within the stability range, from slightly acidic to neutral. This pH condition facilitated the successful formation of the goethite phase, as confirmed by X-ray diffraction (XRD) analysis (details provided in the Appendix A).

The corrosion inhibitor (PIL 03) is considered a compound with intermediate and satisfactory performance. The appearance of different phases of PIL 02 was not observed; its difference was in the size of the carbon chain. Likely, with the increase in the carbon chain length, the material’s surface was better covered with the corrosion inhibitor, resulting in superior protective capacity compared to PIL 02. The corrosion inhibitor PIL 02 was considered the protic ionic liquid tested, which presented the least satisfactory results of all evaluated samples, with weak protective capacity owing to its good solubility in water, but a carbon chain size that did not reach a satisfactory surface coverage.

Thus, this low protective capacity was confirmed by the XRD study (Figure 11), where the phases found were indistinguishable from those in the study without the addition of the inhibitor. In summary, PIL 02 and PIL 03 showed an approximately equal protective capacity, and a difference was observed when the carbon chain length was evaluated, where PIL 02 had three carbons (propionic acid) and PIL 03 had five carbons (pentanoic acid).

### 2.4. Mechanism of Inhibition

A variety of chemical compounds has been employed as corrosion inhibitors (CIs) over the past decade. Among them, ionic liquids, both subdivisions aprotic and protic, have environmental and exclusive interests and are essential to this research. Individual combination of acids and bases, with long alkyl chains, particularly ranging from 8 to 18 carbon atoms, exhibit notable inhibitory properties that stem from their relatively larger molecular size to cover the metal surface after the addition in solution. Undoubtedly, this attribute enhances the distance between the metal surface and the corrosive environment/electrolyte, effectively providing a protective barrier/film for the material [7].

There has been limited research on ionic liquids, both subdivisions, containing shorter alkyl chains, typically with a maximum of six carbon atoms, even though existing data suggest they may be highly effective as corrosion inhibitors. Research on these smaller molecules has shown encouraging results in corrosion reduction; however, studies emphasizing their inhibitory capabilities are still notably scarce in the available literature about corrosion mitigation [81]. 

This article aims to address this issue by examining both the efficiency and mechanisms of these short-chain ionic liquids in the context of their potential applications as corrosion inhibitors. Moreover, a thorough investigation of these mechanisms is essential in corrosion inhibitor studies, as this theoretical comprehension facilitates an understanding of the surface process for those new to corrosion science, mainly graduate students and apprentice researchers.

Therefore, an extensive analysis of gravimetric, electrochemical, and surface results allows researchers to suggest a probable mechanism. It is widely accepted that ionic liquids achieve their inhibitory effect through the adsorption of both cationic and anionic species at the electrochemical interface between the metal and the solution [37,82]. This adsorption process is typically affected by the ionic liquid’s molecular structure and species composition [81], as well as by the surface charge of the metal [83] and the charge distribution within the inhibitor molecule itself [31]. 

In this regard, such perceptions facilitate the comprehension of the inhibition mechanism of protic ionic liquids. Notably, oxidation and reduction reactions (Figure 12) occur simultaneously on the steel surface at different sites. To enhance clarity in comprehending the illustration, both corrosive processes were combined into a single image to provide a didactic/visual representation that effectively conveys the reality of the oxidation-reduction reaction occurring in carbon steel within a saline solution aided by protic ionic liquids.

Initially, the cathodic process begins with the reduction reaction at the designated sites, marking the start of the inhibition process.O_2_ + 2H_2_O + 4e^−^ → 4OH^−^(8)

In the electrochemical reduction of O_2_ (Equation (8)), resulting from an aerated medium, it was effectively reduced, resulting in the production of hydroxyl ions (OH^−^). Production was enhanced by the continuous flow of ions on the metallic surface, which resulted in a remarkable change in magnitude.

Continuing the discussion of the mechanism, free electrons migrate from more active sites to less active, facilitating the reduction of oxygen ions in the solution. These reduced oxygen ions then combine with four free electrons and water molecules to produce hydroxyl ions (OH^−^) altering the electrolytic environment [84]. Additionally, the generated hydroxyl ions are released into the solution and migrate towards the active surface of the cathode.

This ionic movement contributes to the overall electrochemical reactions at the cathode, leading to the corrosion of the entire surface over the time of exposure, which in this work was from 1 to 24 h of evaluation. The reduction of oxygen and subsequent hydroxyl ion formation demonstrate dynamic interactions at the cathodic interface.

As a result, the steel layer is exposed to hydroxyl ions (OH^−^), which interact with the cationic component of the protic ionic liquids, ultimately disrupting the protective mechanism of the inhibitors in the cathodic region, as depicted on the left side of Figure 12. In the anodic region, Fe(s) undergoes dissolution into Fe ions (Fe^2+^) due to the initiation of the oxidation reaction [31].

This dissolution facilitates the chemical attack on the metal surface, which becomes positively charged, thus attracting chloride anions from the solution to the surface. Then, these chloride ions adhere to the surface of the carbon steel, forming a cover that further impacts the corrosion resistance of the system [85].

Due to this attraction, in saline solution, the concentration of Cl^−^ anions on the metallic surface increases drastically, primarily because of the Coulomb forces. This process assists the rapid and efficient adsorption of specific molecules onto the metal surface, including organic compounds. Successively, at this stage, these compounds are predominantly positively charged, a result of charge transfer, and they attract the cationic fraction of protic ionic liquids diluted on the solution, thereby contributing to the inhibition mechanism. These interactions ensure the effectiveness of the corrosion inhibition (PILs) process by enhancing the adsorption of the inhibitor molecules on the metal surface under the influence of electrostatic forces to protect the material [85].

As a result of this interaction, between the ions absorbed on the metal surface and the cationic fraction (+) of the PILs, the coupling between the cationic and anionic fractions of the protic ionic liquids on the steel surface is evident, functioning as a barrier against external factors, including the corrosion process and chloride permeation into the oxide/oxyhydroxide layer of corrosion product [83].

Furthermore, this protective process involves the cationic fraction of the ionic liquid being attracted to Cl^−^ ions adsorbed onto the steel surface, while the anionic fraction is drawn to the positively charged areas formed by Fe^2+^ dissolution into the solution. Importantly, this dual interaction generates a potential difference on the steel surface, effectively inhibiting corrosion and contributing to the stability of the material under corrosive conditions [85].

Over time, the anionic fraction demonstrated a pronounced influence on the metallic substrate, leading to the formation of a protective layer resembling a simplified coating. This outer layer facilitated the adaptation of the cationic fraction, which subsequently interacted with the adsorbed species, completing the ionic liquid’s protective coverage on the metallic surface. The results of this study indicate that all the Protic Ionic Liquids (PILs) tested exhibit promising potential as corrosion inhibitors, achieving efficiencies ranging from 50% to 75%. Notably, these compounds are distinguished by their inherent sustainability and non-bioaccumulative nature. 

The PILs evaluated primarily acted as mixed-type and anodic adsorption inhibitors. In specific scenarios, a predominantly anodic inhibition mechanism was observed, which induced passive behavior at higher concentrations. For instance, in the case of PIL A, the adsorbed layer effectively impeded the aggressive action of the electrolyte solution on the steel substrate, thereby mitigating corrosion. This behavior underscores the ability of PILs to enhance surface protection through the formation of robust adsorbed films, contributing to their effectiveness as environmentally friendly alternatives for corrosion inhibition in saline environments (Figure 12).

## 3. Materials and Methods

### 3.1. Materials

Initially, for the reliable and precise reproduction of the experiments, the components utilized in the synthesis were purchased by Aldrich (Millipore Sigma, St. Louis, MO, USA), with a mass purity of 0.99. At the same time, the acids were acquired from Sigma (Millipore Sigma, St. Louis, MO, USA) (mass purity of 0.996). Formerly, the protic ionic liquids (PILs) were produced by a neutralization reaction of an equimolar mixture of mono-ethanolamine (mass purity ≥ 99.0 wt. %) combined with formic acid (mass purity ≥ 000 wt. %), propanoic acid (mass purity 0000 wt. %), or pentanoic acid (mass purity 0000 wt. %) [86,87,88,89,90].

### 3.2. Protic Ionic Liquid Synthesis

A glass triple-necked flask equipped with a thermometer to evaluate reaction temperature, a reflux condenser, and PIL were synthesized with a dropping funnel. In summary, the chemical nomenclature 2-Hydroxy ethanolamine Formate (PIL 01 = 2-HEAF), 2-Hydroxy Ethanolamine Propionate (PIL 02 = 2-HEAP), and 2-Hydroxy Diethanolamine Pentanoate (PIL 03 = 2-HEAPe) were formed, succeeding in an acid–base reaction represented in Figure 13 [86,87,90,91]. In detail, an equimolar amount of formic, propanoic, and pentanoic acids were added drop by drop to the flask in which a base was placed, stirring with a magnetic bar at a constant speed (300 RPM).

Notably, the flask was mounted in an ice bath to avoid overheating the system during the reaction, keeping the process stable. Then, after the reaction, the product was dried for approximately 48 h under a vacuum (20 kPa) to remove humidity, and it was then stored in a dark bottle to avoid luminous influence [91]. The Karl Fischer method was chosen to determine the water content in the system. Protic ionic liquid chemical structures were confirmed by ^1^H NMR, ^13^C NMR, FTIR, and TGA, detailed in Appendix A. It is essential to mention that Rodrigues and co-authors discovered a similar structure in 2018 [87].

### 3.3. Sample Preparation for General Evaluations

The chemical composition of carbon steel used in the experiments (wt. %) was with A36 and with the assistance of PDA 7000 Optical Emission Spectrometer (Shimadzu Corporation, Kyoto, Japan), with average C = 0.21029, Si = 0.03306, Mn = 0.50905, P = 0.00569, S = 0.00841, Ni = 0.02425, and Cr = 0.0233, ratifying the legibility of the material. Initially, the steel samples (A36) with dimensions of one cm^2^ were sanded with every grit paper 120, 220, 400, 600, and 1200, without further polishing. At that point, water, ethanol, and acetone were selected to wash the material before the electrochemical evaluation. After each washing step, samples were dried with hot air [92].

### 3.4. Corrosion Tests

#### 3.4.1. Weight Loss and Immersion Measurements

The carbon steel samples of ASTM A36 (1.0 × 3.5 × 0.5 cm) were primed with the process (abraded, rinsed, and dried) until the first weight (*W*_1_). The immersion test was performed in a volume of 500 mL of 3.5% wt. of sodium chloride (NaCl), calculated by the total sample area without or with 1000 ppm PIL concentration, with a specific period of 100 h. Subsequently, the corrosion products were eliminated by Clark solution, and the samples were dried and weighed again (*W*_2_). Each experiment was conducted in triplicate. The weight loss (Δ*W*) was calculated by the following Equations (9) and (10), the same as in the literature about mass-loss measurements [3,16,93,94,95]:(9)$$W=W_{1}-W_{2}$$

The corrosion rate, denominated by ν_corr_ (mm/y), was calculated by Equation (10), as follows:(10)$$\nu_{corr}=\frac{∆W\;\cdot \;\;k}{S\cdot t\cdot p}$$

Fundamentally, the symbol (*S*) is the sample area (cm^2^), (*t*) is the dousing period (h), (*p*) is the density (g/cm^3^), and the constant (*k*) is related to the selected time unit (mm/y) according to ASTM G1 [96]. The inhibition efficiency, %*IE*, was calculated by the following:(11)$$\%\;IE=\frac{\nu_{corr}\left(b\right)-\nu_{corr}(i)}{\nu_{corr}\left(b\right)}\times 100$$
where ν_corr(i)_ and ν_corr(b)_ are the corrosion rates with and without inhibitors.

#### 3.4.2. Electrochemical and Conductivity Measurements

For the electrochemical measurements, a conventional three-electrode cell arrangement with a working electrode of A36 carbon steel (0.5 cm × 0.5 cm), an Ag/AgCl wire as a reference electrode, and a platinum wire counter electrode (1.2 cm × 1.2 cm) was used with the use of an Autolab 302N potentiostat (Metrohm Autolab, Utrecht, The Netherlands) [37,97]. In addition, the carbon steel samples (W.E) were immersed in solutions with PILs in increasing concentrations of 250, 500, and 1000 ppm for the electrochemical evaluation in saline electrolyte (NaCl 3.5 wt. %) [92,98,99,100]. The electrochemical impedance spectroscopy technique was tested with an applied AC-perturbation signal of 10 mV in a frequency range from 100 kHz to 0.01 Hz. For the measurement of polarization curves, the sweep rate was 1 mV/s. Therefore, each recording was repeated at least three times to investigate the accuracy of the system [37,97].

The critical micelle concentration (CMC) of protic ionic liquids was determined by plotting the conductivity data against the (PILs) concentration at 25 °C. The solution was added dropwise using a micropipette (5 µL), assisted by magnetic stirring (200 RPM) to dissolve the electrolyte. The CMC value was determined by the time when the solute addition did not change the conductivity value during a certain period (stability phase) [32,60].

### 3.5. Protic Ionic Liquids Characterization and Test Methods

Each mixture was prepared with a known mass of protic ionic liquid and solvent, and both were injected into a glass vial using a syringe. The mixtures were sealed with an aluminum cap and rubber plug in the vials. In addition, the space in the vials was minimized to avoid evaporation loss. The mixtures were injected into a DSA 5000 densimeter, and simultaneous measurements of the density and speed of sound at each temperature and atmospheric pressure of 100 kPa were obtained [91].

DSA 5000 (Anton Par, Graz, Austria) was used to evaluate the density and sound velocity values of pure liquids. The density (ρ) and viscosity (η) were measured at atmospheric pressure (101.325 kPa) in the temperature range of T = (293.15–333.15) K, using an Anton Paar SVM 3000 digital oscillation U-tube, with the following standard uncertainties: u: u(x_1_) = 0.003; u(T) = 0.01 K; u(η) = 0.02∙η mPa∙s; and u(ρ) = 0.0015 g∙cm^−3^. The speed of sound was measured with the following standard uncertainties: u: u (x 1) = 0.0008; u (T) = 0.01 K; u(v) = 0.9 m. s^−1^. The ^1^H and ^13^C NMR experiments were performed using a Bruker Avance DRX 500 spectrometer (Bruker, Billerica, MA, USA) operating at 500 and 125 MHz for ^1^H and ^13^C, correspondingly equipped with an inverse detection One Probe.

In addition, to assess the formation of the protic ionic liquid, PILs (30 mg) were dissolved in 0.5 mL of deuterium oxide (D_2_O) (99.9% D/Sigma), and the residual solvent peak was used as an internal reference (4.81 ppm) and analyzed in 5 mm tubes (Wilmad, Vineland, NJ, USA).

For this examination, an infrared spectrometer (Fourier Transform Cary 630, Agilent Technologies, Santa Clara, CA, USA) was used, which enabled the evaluation of liquid and solid samples. Thus, the samples were applied directly to the spectrometer without prior preparation to avoid error. Absorbance spectra were collected at a wavelength in the range of the most significant interest for organic components (400–4000 cm^−1^) and with a spectral resolution of 1 cm^−1^. TGA analysis was performed for all protic ionic liquids to investigate the thermal stability of the materials under different atmospheric conditions using SDTA 851 (Mettler Toledo, Zurich, Switzerland). All the PILs were similar under an N_2_ atmosphere at 30–400 °C, with a flux of 10 °C per 50 mL. Finally, detailed information (tables and figures) is provided in the Appendix A.

### 3.6. Surface Analyses

The surface evaluation was possible with the aid of optical microscopy. The surface of the sample was evaluated as a saline solution containing 1000 ppm of all PILs evaluated for 24, 48, and 72 h. In addition, to promote a detailed evaluation of the material, Scanning Electron Microscopy (SEM) was applied using the equipment Quanta 450-FEG Thermo Fisher Scientific, Hillsboro, OR, USA). AFM—Atomic Force Microscopy—was utilized to study the changes in the surface morphology of A36 carbon steel after 24 h of immersion at 298 K for surface analysis. The AFM measurements were obtained in the intermittent contact mode with an Asylum MFP-3D BIO equipment (Oxford Instruments, Santa Barbara, CA, USA) along with curved radius tips smaller than 10 nm and a resonant frequency of 75 kHz. The scan area of AFM imaging is at least 10 µm × 10 µm.

Finally, the corrosion products (oxides and hydroxides) were evaluated by applying the X-ray diffraction method, where steel (A36) was immersed for 30 days with a fixed inhibitor concentration (1000 ppm) in the absence of an inhibitor. After this period, the former corrosion product was filtered and evaluated using an X-ray diffractometer (RX/DMAXB, Rigaku, Tokyo, Japan).

## 4. Conclusions

From the current investigation, the following primary conclusions may be drawn: PIL 01 (2-HEAF), PIL 02 (2-HEAP), and PIL 03 (2-HEAPe) demonstrated significant corrosion inhibition on carbon steel (A36) in a 3.5 wt. % NaCl solution at room temperature (25 °C). The corrosion inhibition efficiency was in the following order: PIL 01 (2-HEAF) > PIL 02 (2-HEAP) > PIL 03 (2-HEAPe). Mass loss tests enabled the development of an adsorption isotherm model, with all PILs being classified accordingly. The Langmuir model best represented the steel/solution interface. Based on the Arrhenius and transition state equations, the dissolution of compounds in the saline solution was found to be an endothermic reaction, with ΔG_0ads_ values of −22.89 kJ mol−1 (PIL 01), −22.98 kJ mol−1 (PIL 02), and −20.67 kJ mol−1 (PIL 03).

The steady-state open-circuit potential (E_ocp_) was utilized to analyze the chemical interactions between the PILs and the electrolyte. The E_ocp_ results indicated a marked increase in the potential values for the PILs, with more positive values compared to the blank, as a function of time. Polarization curves showed that all PILs effectively suppressed both the cathodic and anodic reactions of the A36 electrode. PIL 01 exhibited mixed-type inhibition characteristics, while PIL 02 and PIL 03 displayed anodic-type inhibition behavior.

Electrochemical impedance spectroscopy (EIS) revealed the superior performance of PIL 01 over the other two PILs. The efficiency of PIL 01 increased with concentrations up to 1000 ppm, whereas PILs 02 and 03 exhibited reduced effectiveness with higher concentrations, possibly due to the increasing carbonic acid chain length that diminished inhibition efficiency. Morphological techniques, including optical microscopy, scanning electron microscopy (SEM), and atomic force microscopy (AFM), confirm the protective effect of PIL 01 on the A36 material, with results consistent with electrochemical data. These techniques proved essential in understanding the mechanisms behind corrosion inhibition.

X-ray diffraction (XRD) was employed to analyze the oxides formed on the steel surface. The XRD results confirmed the adsorption mechanism (physisorption) and provided valuable insights into the inhibition mechanism, revealing phases that are characteristic of systems with effective corrosion inhibitors in the reaction medium.

## Figures and Tables

**Figure 1 molecules-30-01033-f001:**
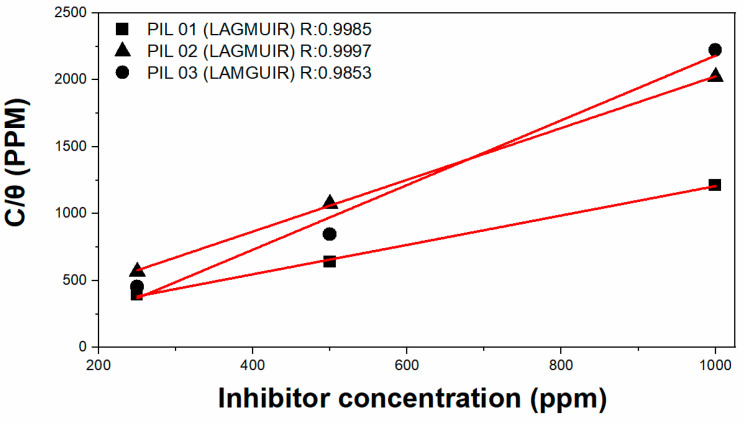
Langmuir isotherms for the adsorption of all PILs on the carbon steel in the NaCl electrolyte.

**Figure 2 molecules-30-01033-f002:**
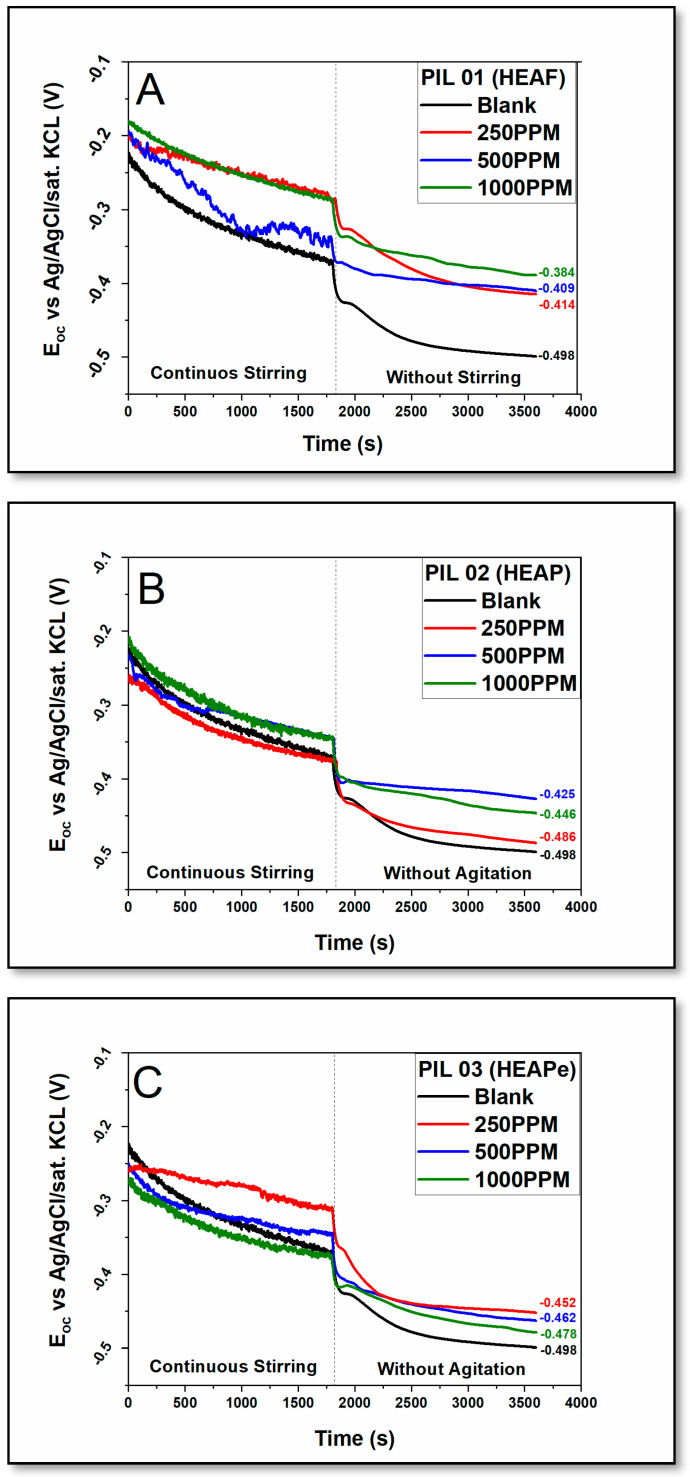
Differences in the Open Circuit Potential (OCP) with an interval for the A36 carbon steel samples in 3.5% NaCl electrolyte in the existence and absence of the protic ionic liquids ((**A**) = PIL 01; (**B**) = PIL 02; and (**C**) = PIL 03).

**Figure 3 molecules-30-01033-f003:**
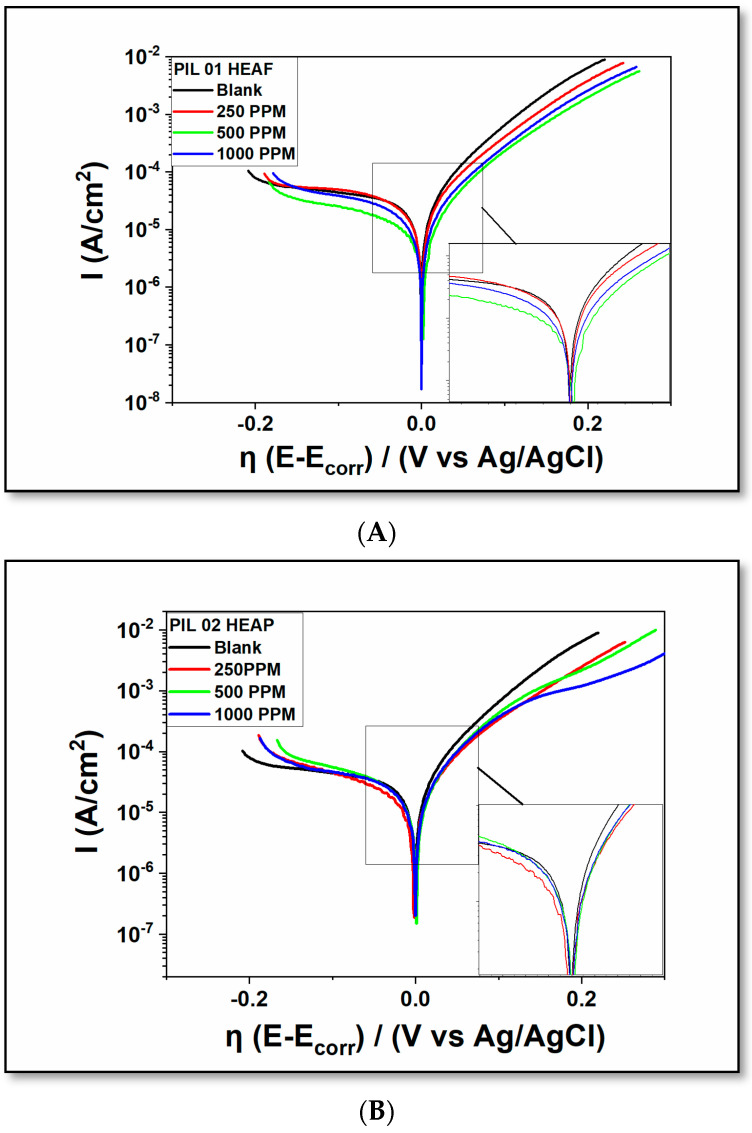
Overpotential plots of the potentiodynamic polarization curves for PIL 01–03 ((**A**) = PIL 01; (**B**) = PIL 02; and (**C**) = PIL 03).

**Figure 4 molecules-30-01033-f004:**
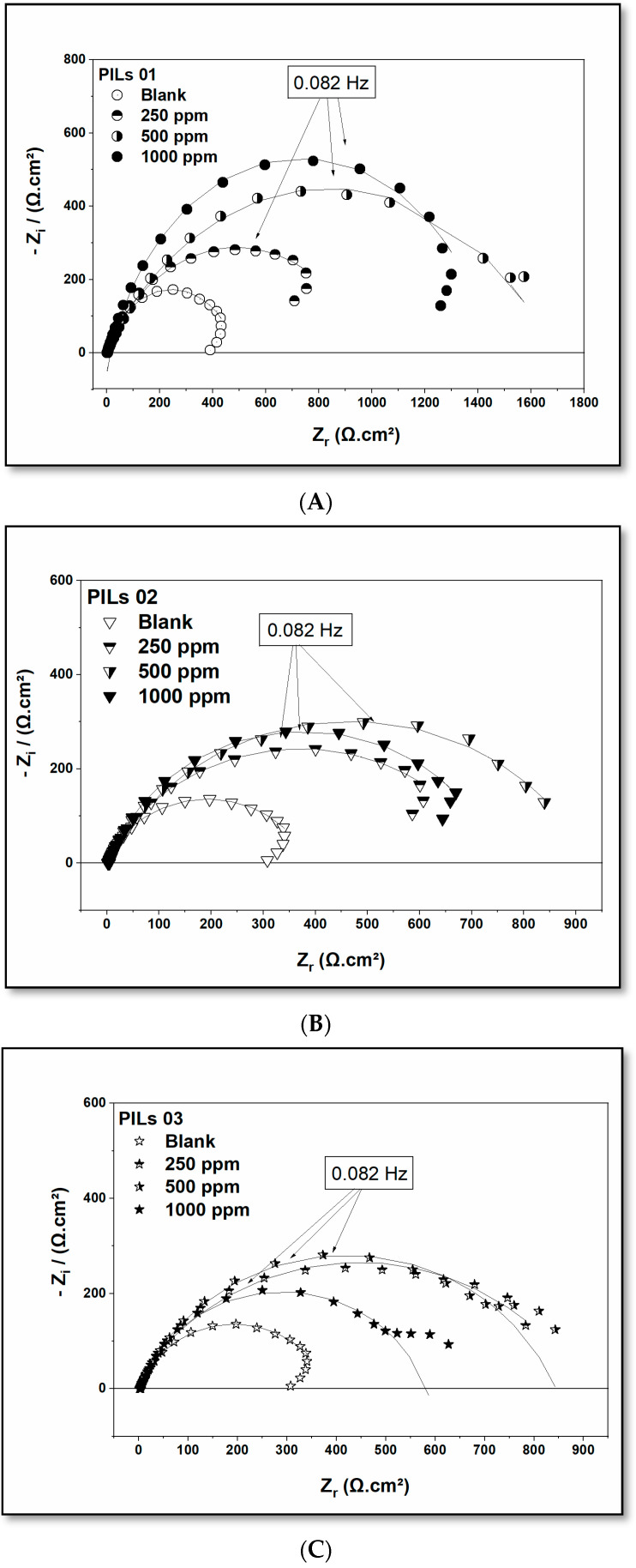
Plots for the impedance of carbon steel (A36) in 3.5% NaCl in the presence and absence of PILs ((**A**) = PIL 01; (**B**) = PIL 02; and (**C**) = PIL 03) with different concentrations.

**Figure 5 molecules-30-01033-f005:**
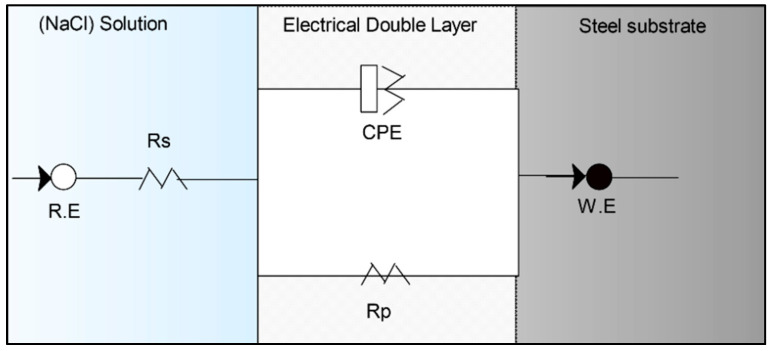
Equivalent circuit modes for the carbon steel electrode (ASTM A36).

**Figure 6 molecules-30-01033-f006:**
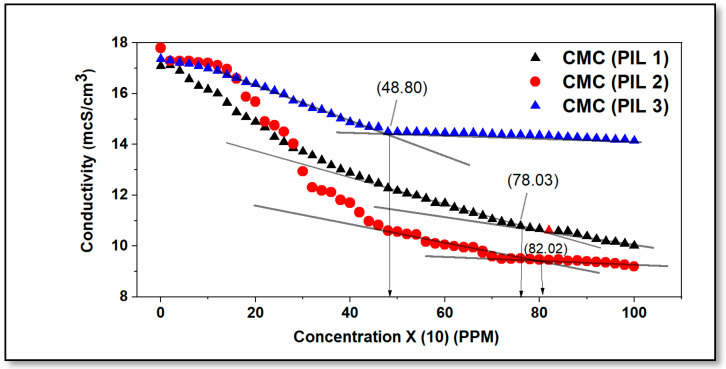
Variation in conductivity with inhibitor concentration for the carbon steel electrode in 3.5% NaCl solution at 25 °C.

**Figure 7 molecules-30-01033-f007:**
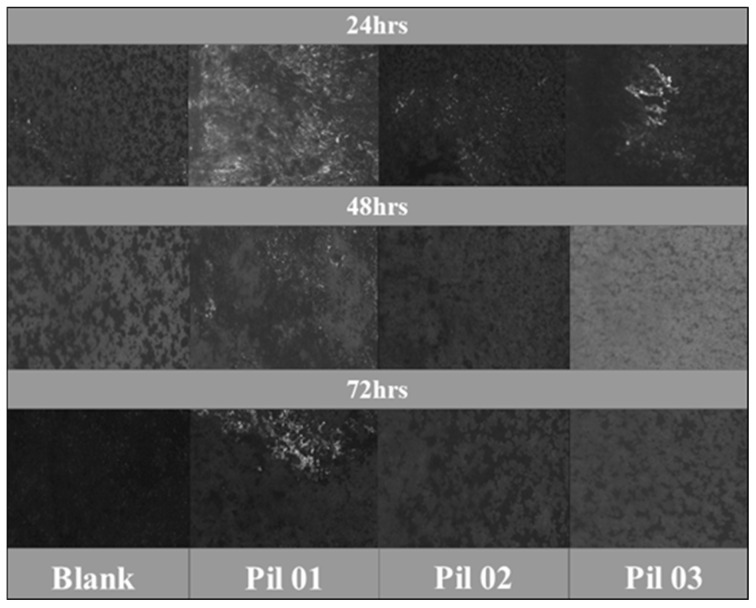
Photographs of the ASTM A36 carbon steel surfaces after 24, 48, and 72 h of immersion without inhibitor and in the presence of 500 ppm of each inhibitor—PIL 01 (2-HEAF), PIL 02 (2-HEAP), and PIL 03 (2-HEAPe).

**Figure 8 molecules-30-01033-f008:**
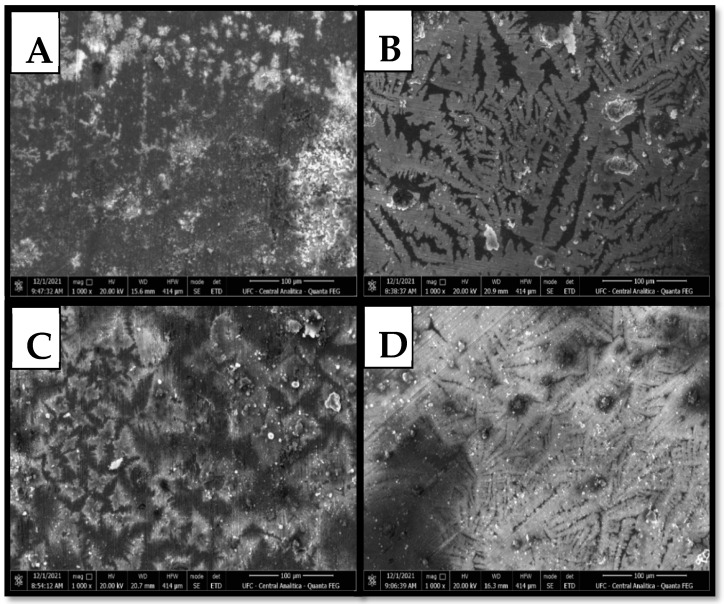
SEM micrographs (1000×) of carbon steel: (**A**) absence of PILs and in the presence of 1000 ppm of (**B**) PIL 01 (HEAF), (**C**) PIL 02 (HEAP), and (**D**) PIL 03 (HEAPe) after polarization tests at 25 °C.

**Figure 9 molecules-30-01033-f009:**
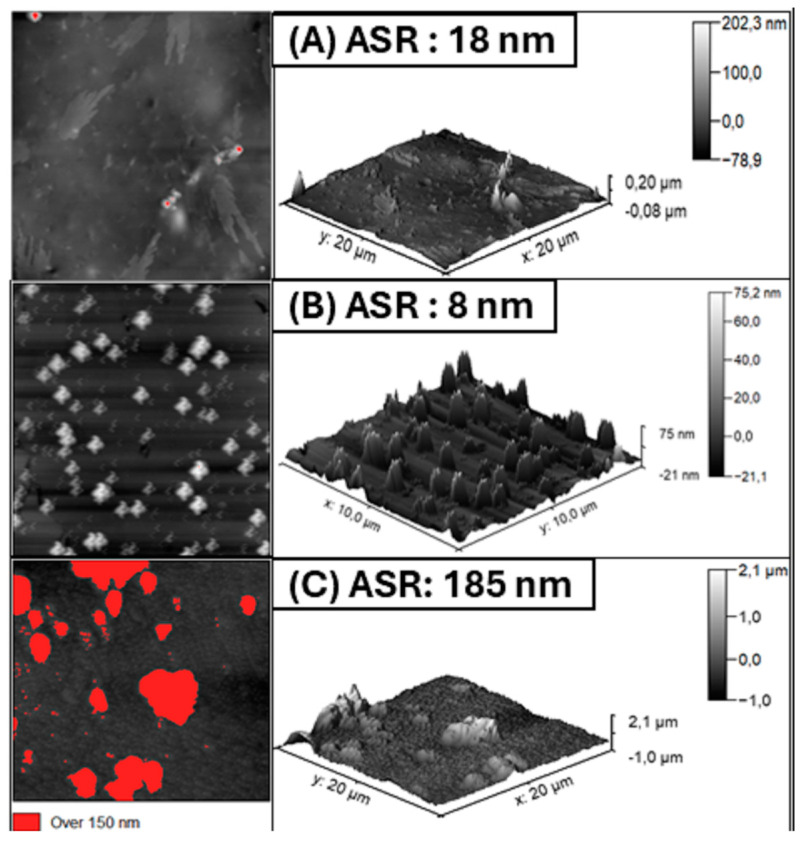
AFM phase (**left**) and AFM 3D (**right**) images of the A36 alloy surface after immersion in 3.5 wt. % NaCl: (**A**) PIL 01 (1000 PPM), (**B**) Polished Control (P.C), and (**C**) Blank NaCl 3.5%. ASR—Average Surface Roughness.

**Figure 10 molecules-30-01033-f010:**
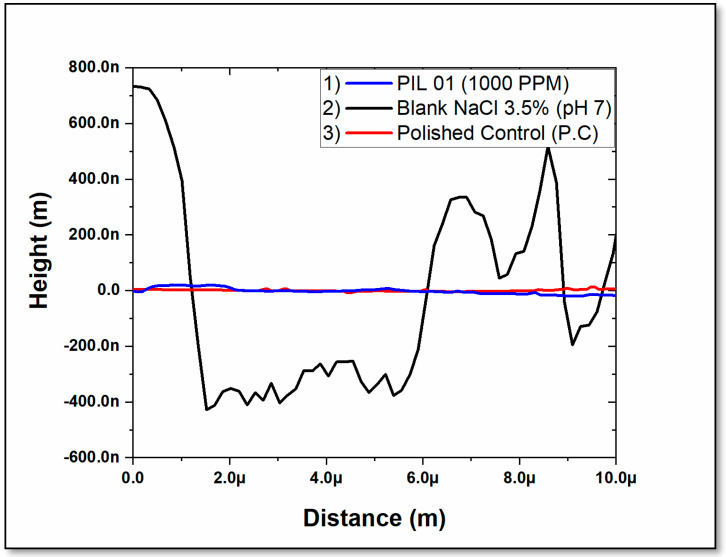
Two-dimensional roughness profile of AFM images: (1) PIL 01 (1000 PPM), (2) Blank NaCl wt. 3.5%. and (3) Polished Control (P.C).

**Figure 11 molecules-30-01033-f011:**
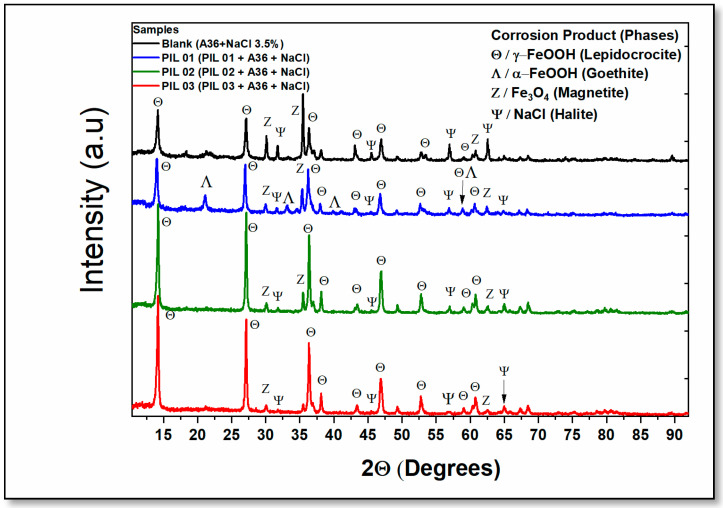
X-ray diffraction (XRD) patterns of rust products in the absence (blank) and presence of 1000 ppm of three different protic ionic liquids (PIL 01–03).

**Figure 12 molecules-30-01033-f012:**
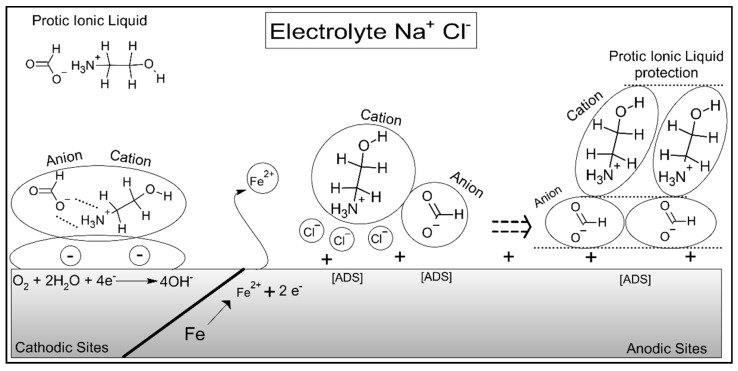
Mechanism of the corrosion inhibition of carbon steel (A36) with LIPs in NaCl 3.5% (anodic and cathodic sites).

**Figure 13 molecules-30-01033-f013:**
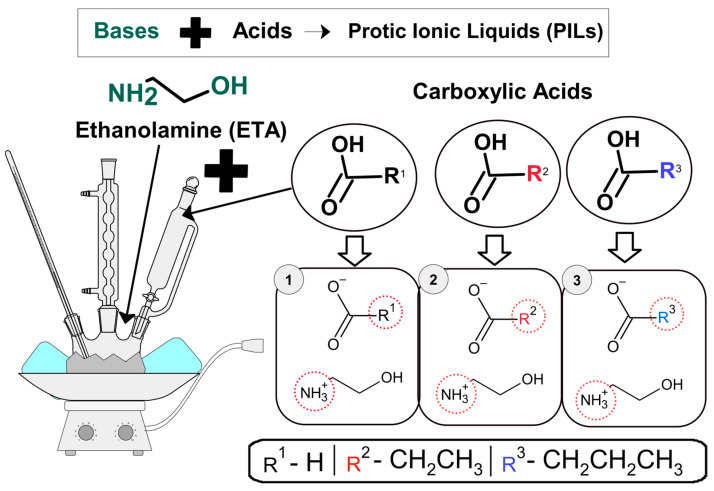
Schematic route for PIL synthesis.

**Table 1 molecules-30-01033-t001:** Weight loss corrosion parameters of carbon steel immersed in 3.5% NaCl solution without and with 250, 500, and 1000 ppm of different protic ionic liquids for 100 h.

	250 ppm			500 ppm			1000 ppm		
PIL	A (ΔW) ± SD	νcorr (mm/y)	%IE	A (ΔW) ± SD	νcorr (mm/y)	%IE	A (ΔW) ± SD	νcorr (mm/y)	% IE
Blank	0.07 ± 0.01	1.31	0	0.07 ± 0.01	1.3141	0	0.07 ± 0.01	1.3141	-
PIL 01	0.02 ± 0.006	0.47	63.5	0.01 ± 0.005	0.2840	78.3	0.01 ± 0.03	0.22	82.6
PIL 02	0.04 ± 0.02	0.73	44.1	0.03 ± 0.009	0.7022	46.5	0.03 ± 0.01	0.66	49.4
PIL 03	0.03 ± 0.01	0.58	55.1	0.02 ± 0.0005	0.5358	59.2	0.03 ± 0.02	0.72	44.9

A—average (ΔW); νcorr—corrosion rate; SD—standard deviation.

**Table 2 molecules-30-01033-t002:** Polarization parameters of A36 Carbon Steel in 3.5% NaCl without and with different concentrations of PIL 01 (2-HEAF), PIL 02 (2-HEAP), and PIL 03 (2-HEAPe).

	Blank	PIL 01			PIL 02			PIL 03		
		250	500	1000	250	500	1000	250	500	1000
E_corr_	−0.564 ± 0.030	−0.438 ±0.002	0.410 ± 0.01	−0.375 ± 0.013	−0.526 ± 0.012	−0.480 ± 0.013	−0.463 ± 0.037	−0.477 ± 0.009	−0.452 ± 0.009	−0.566 ± 0.011
I_corr_	9.47 ± 0.30	4.18 ± 0.40	2.50 ± 0.21	1.57 ± 0.80	5.86 ± 0.70	5.61 ± 0.10	5.24 ± 0.31	5.35 ± 0.5	4.32 ± 0.14	5.97 ± 0.21
Eff.		60.4	71.9	80.1	40.4	47.3	52.5	50.1	60.7	32.6

Units: E_corr_ (V), I_corr_ (_µA/cm_^2^); Eff.—efficiency in percentage (%).

**Table 3 molecules-30-01033-t003:** Impedance factors of A36 in 3.5% wt. NaCl with and without different NaCl concentrations.

PILs (ppm)	Rs (Ω·cm^2^)	CPE		Cdl (F·cm^−2^) × 10^−3^	Rp (Ω·cm^2^)	I.E (%)
		Y_0_(s^n^·Ω^−1^·cm^−2^) × 10^−3^	n. × 10^−2^			
Blank	1.86 ± 0.08	2.41 ± 8.37	75 ± 2	2.67 ± 1.31	485 ± 29	----
01–250	4.06 ± 0.10	5.88 ± 4.51	75 ± 3	1.15 ± 1.01	1224 ± 91	60.1 ± 2.83
01–500	5.12 ± 0.14	1.11 ± 1.80	73 ± 1	1.37 ± 2.75	1632 ± 78	70.2 ± 1.39
01–1000	6.47 ± 0.31	0.52 ± 3.36	70 ± 3	0.56 ± 5.51	2474 ± 98	80.3 ± 0.86
02–250	3.01 ± 0.27	1.63 ± 1.32	74 ± 1	2.11 ± 2.09	848 ± 46	42.5 ± 3.25
02–500	3.61 ± 0.27	2.06 ± 0.20	76 ± 2	2.57 ± 1.02	901 ± 67	45.8 ± 4.25
02–1000	3.88 ± 0.21	1.34 ± 0.21	74 ± 2	1.48 ± 3.09	996 ± 26	51.2 ± 1.31
03–250	3.16 ± 0.14	2.68 ± 1.09	75 ± 1	4.02 ± 2.13	1051 ± 78	52.7 ± 3.25
03–500	4.23 ± 0.32	1.95 ± 2.42	75 ± 1	2.56 ± 5.11	1156 ± 30	58.0 ± 1.13
03–1000	3.18 ± 0.12	3.18 ± 3.52	76 ± 1	4.21 ± 0.89	789 ± 23	37.0 ± 1.12

**Table 4 molecules-30-01033-t004:** X-ray patterns of rust product phases (oxides and hydroxides).

Samples	Mineral	Oxide	Hydroxide	
Blank	Halite (NaCl)	Magnetite (Fe_3_O_4_)	Lepidocrocite (γ-FeOOH)	-
01	Halite (NaCl)	Magnetite (Fe_3_O_4_)	Lepidocrocite (γ-FeOOH)	Goethite (α-FeOOH)
02	Halite (NaCl)	Magnetite (Fe_3_O_4_)	Lepidocrocite (γ-FeOOH)	-
03	Halite (NaCl)	Magnetite (Fe_3_O_4_)	Lepidocrocite (γ-FeOOH)	

## Data Availability

The original contributions presented in the study are included in the article/Appendix A, further inquiries can be directed to the corresponding authors.

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
