# Peer review of "Assessment of Sustainable Ethanolamine-Based Protic Ionic Liquids with Varied Carboxylic Acid Chains as Corrosion Inhibitors for Carbon Steel in Saline Environments"

_molecules, 2025, doi:10.3390/molecules30051033_

Round 1
Reviewer 1 Report (Previous Reviewer 2)
Comments and Suggestions for Authors
The paper I am reviewing is the revised by the authors manuscript with the number “molecules-3385561”.
In the updated version, the authors have addressed all my comments and recommendations and answered my questions that I asked during the initial review.
After revision, the content became more comprehensive, which greatly improved the overall quality. I believe that the article has reached the standard of publication.
I believe that the manuscript can be accepted after a few minor corrections:
1) The Microsoft Word template that the authors used to prepare the manuscript is inappropriate. It is true that the logo of Molecules is visible on the first page, but in the header and footer of each page there is a reference to Metals ("Metals 2021, 11, x FOR PEER REVIEW").
2) line 370
Figure 4 shows that the polarization curves were measured from -0.2 V to 0.2 V, not from -0.2 V to 2 V, as written in line 370.
3) Figure 10
The 3D drawing at the bottom of figure 10 should be labeled: (C) ASR: 185 nm, not: (A) ASR: 185 nm.
4) line 874
"In the electrochemical reduction of O2 (equation 10)"
It should be written: "In the electrochemical reduction of O2 (equation 11)"
Author Response
Response to Reviewer 1
1. Summary
Dear reviewer, thank you for your excellent feedback and for examining this manuscript about sustainable corrosion inhibitors. Moreover, please find our detailed responses to your comments below. The revisions and corrections have been incorporated and are highlighted in the re-submitted files using track changes.
We sincerely apologize for any grammatical inaccuracies in the article or any other aspect of the material. In addition, we deeply appreciate your understanding and the time you have taken to provide your feedback.
Thank you for your patience and consideration.
3. Point-by-point response to Comments and Suggestions for Authors
Comments 1: The Microsoft Word template that the authors used to prepare the manuscript is inappropriate. The logo of Molecules is indeed visible on the first page, but in the header and footer of each page, there is a reference to Metals ("Metals 2021, 11, x FOR PEER REVIEW").
Response 1:
Thank you for your thorough evaluation of our manuscript and your valuable comments. We appreciate your feedback; we appreciate your feedback. As stated, we have already implemented the correction and included the accurate journal name (Molecules).
Comments 2: Figure 4 shows that the polarization curves were measured from -0.2 V to 0.2 V, not from -0.2 V to 2 V, as written in line 370.
Response 2:
Thank you for your thorough evaluation of our manuscript and your valuable comments. We appreciate your feedback; The modification has been made.
Comments 3: The 3D drawing at the bottom of Figure 10 should be labeled: (C) ASR: 185 nm, not: (A) ASR: 185 nm
Response 3:
Thank you for your thorough evaluation of our manuscript and your valuable comments. We appreciate your feedback; The modification has been made in Figure 10.
Comments 4: In the electrochemical reduction of O (equation 10)" It should be written: "In the electrochemical reduction of O (equation 11)"
Response 4:
Thank you for your thorough evaluation of our manuscript and your valuable comments. We appreciate your feedback; The modification has been made.
4. Response to Comments on the Quality of English Language:
We have no further considerations
5. Additional clarifications
We have no further considerations

Reviewer 2 Report (Previous Reviewer 3)
Comments and Suggestions for Authors
Among all the corrections requested to improve the manuscript, only the underlined ones were carried out, in the resubmitted manuscript, the rest were not taken into consideration.
1. English needs to be amended. There are some grammatical and typos in the text.
2. The introduction needs to be improved. The authors must state the novelty and state-of-the-art about the protic ionic liquids.
3. “K” in equation 2 is not specified.
4. Check line 155, “Where Ê‹corr(b) and Ê‹corr(i) are the corrosion rates with inhibitors and without them”. In this way, the efficiency will be negative.
5. Correct line 235, “A - Average / W - Weight (1) and (2) / Ê‹corr - Corrosion Rate / SD - Standard Deviation”. This sentence in the caption is not necessary
6. The X and Y axes in the Nyquist plot must be the same. You must also report Bode plots, together with the Nyquist, to prove how many time constants you have.
7. The results reported in Table 3 don’t seem to agree with the plot in Figure 5. For instance, the Rp for PILs01 and PILs02 1000ppm seems lower than 500ppm.
8. You must also report the X2 in Table 3.
9. Line 434, ”Specifically, to calculate inhibition efficiency (I.E) and double-layer capacitance (Cdl), Eqs. (4)–(5) were employed, respectively, where Rp and Rp(inh) represent the polarization resistances in the absence and presence of inhibitors”. Eqs. (8)–(9)
10. Line 470, “Moreover, the EIS results show that the Cdl values decrease with increasing Rs”. Rp no Rs. Besides, based on question 5, this sentence must be rewritten.
11. Why did you specify the dielectric constant (ε) and the protective film thickness (T), with the letters? You didn’t report Helmholtz's equation.
12. The optical microscopy images reported in Figure 8 must be changed. You can’t distinguish anything with these images.
13. Line 596, “In contrast, it is observed that a significant reduction in surface damage was observed, with a more uniform appearance and less evidence of corrosive damage”. From the look of the SEM images, it seems that in the presence of the inhibitor, the surface looks etched (Figures B and D); you can see the structure of the metal.
14. The AFM images need to be improved. They lack a proper scale, and they do not seem to be well prepared. Something is present on the polished sample surface, likely some unremoved polished materials
15. Line 647, “As shown in Figure 10, there is A apparent increase in both the roughness and thickness of the carbon steel surface after 24 hours of EXPOSION in saline electrolyte”. How can you see the thickness from these measurements? Plus, correct the English in this sentence.
16. The XRD part is confusing and needs to be rewritten. The author should focus more on the obtained results and explain why they have this type of oxide without mentioning things that are not even related to this study (e.g., line 779, “Still on the modifications of the corrosion products formed on the metal surface, typically occurs under controlled conditions, such as specific temperature ranges (27 to 70°C) and pH values (7.5 to 9.0), because the modification of oxides at alkaline pH and high temperatures occurs due to the higher concentration of OH- ions and the increase in thermal energy, which accelerates oxidation).
17. Line 729, “Interestingly, the phases identified in the blank sample also appeared in the spectrum of PIL 01 (Fig. 12), although with a notable difference: the presence of the goethite phase in the spectrum of PIL 01”. the peaks related to goethite are also present in the black spectrum.
18. Line 754, “Fuente (2011) indicates that magnetite normally forms a corrosion layer over the surface offering tiniest protection on steel surfaces [89]. Still, countless researchers have emphasized the protective properties of goethite and ferrihydrite concerning metallic surfaces; however, in this paper experiments involving Protic Ionic Liquids (PILs) revealed only the presence of goethite …”. REVEALED ONLY THE PRESENCE OF GOETHITE, this sentence is confusing, you have all types of oxides.
Author Response
1. Summary
First, I would like to express my heartfelt gratitude for your positive and thoughtful review of our manuscript on corrosion inhibitors. Your detailed feedback and recognition of our work is highly appreciated. In response to your comments, we have meticulously revised the manuscript, ensuring that all suggested corrections and improvements have been incorporated. These changes have been marked in the re-uploaded files using the track changes feature for ease of reference. Regarding your observations on the effects of temperature and time, we opt to clarify that these results have been previously published by the corresponding author (Caio Victor Pereira Pascoal) in a separate study. The relevant results can be accessed in the article published in 2024, available at the following DOI: https://doi.org/10.1590/1980-5373-MR-2024-0047.
In sum, we believe that these preliminary results (molecules-3385561) can truly contribute valuable insight to the scarce literature on corrosion inhibitors using protic ionic liquids. We also sincerely apologize for any grammatical errors, omissions, or issues such as low-quality images or inaccuracies in equations in the originally submitted version. Each of these aspects has been thoroughly reviewed to ensure the revised manuscript adheres to the highest standards of clarity, accuracy, and quality. Once again, we deeply appreciate your constructive feedback and encouragement. Receiving such positive and supportive comments in academia is a rarity, and we are truly grateful for your recognition of our efforts and the significance of our work.
Thank you for your time and dedication in reviewing our manuscript.
3. Point-by-point response to Comments and Suggestions for Authors
Comments 1: English needs to be amended. There are some grammatical and typos in the text.
Response 1: Thank you for pointing this information out. We agree with this comment and will improve the topics where the writing/communication was not clear enough to understand the context of the article/text.
Comments 2: The introduction needs to be improved. The authors must state the novelty and state-of-the-art about the protic ionic liquids.
Response 2: Thank you for pointing this information out. We agree with this comment. We will modify the introduction to highlight the novelty of using protic ionic liquids; In this regard, Protic Ionic Liquids (PILs) are a significant advancement in the field of corrosion inhibitors, providing benefits due to their distinctive properties and environmentally friendly nature. In detail, in contrast to conventional inhibitors, which frequently rely on hazardous metals and toxic solvents, PILs are synthesized through candid and economical processes without the utilization of harmful solvents, rendering them an attractive and sustainable possibility [16–24]. In addition, their highly adaptable chemical structure aids precise modification to enhance desirable characteristics. This versatility, in conjunction with their biodegradability and minimal environmental impact, positions PILs as an excellent solution to address the increasing industrial demand for efficient, sustainable, and multifunctional corrosion inhibitors [26] (Page – 1 and 2/32; Line – 75 to 85). We also added this information about the novelty; The utilization of protic ionic liquids (PILs) as corrosion inhibitors for ASTM A36 carbon steel in saline environments represents an innovative and promising approach that addresses significant gaps in the extant corrosion literature. Indeed, by focusing on this material, which is extensively employed in structural and industrial contexts, this study addresses the paucity of research on its corrosion behavior under aggressive conditions. The application of PILs introduces a novel protective mechanism that enhances the durability of the material while aligning with the principles of sustainable material science and metallurgical engineering. This research not only contributes to the understanding of corrosion mechanisms in ASTM A36 but also elucidates the potential of this option, the PILs, as a transformative solution for mitigating material degradation in diverse industrial applications. (Page – 2 and 3/32; Line – 95 to 106).
Comments 3: “K” in equation 2 is not specified
Response 3: Thank you for pointing this information out. We agree with this comment. We add this information, density (g/cm3), and constant (k) which is related to the time unit (mm/y) according to ASTM G1. (Page – 4/32; Line – 151 and 152).
Comments 4: Check line 155, “Where Ê‹corr(b) and Ê‹corr(i) are the corrosion rates with inhibitors and without them”. In this way, efficiency will be negative.
Response 4: Agree. We have, accordingly, changed for Ê‹corr(i) and Ê‹corr(b) after equation 3; (Page – 4/32; Line – 154).
Comments 5: Correct line 235, “A - Average / W - Weight (1) and (2) / Ê‹corr - Corrosion Rate / SD - Standard Deviation”. This sentence in the caption is not necessary.
Response 5: Agree. We have, accordingly, changed for lowercase letters; (Page – 6/32; Line – 257).
Comments 6: The X and Y axes in the Nyquist plot must be the same. You must also report Bode plots, together with the Nyquist, to prove how many time constants you have.
Response 6: Thank you for pointing this information out. We agree with this comment. We agreed on the dimensioning of the X- and Y-axes; however, as there were arcs with different dimensions, it was decided to use a format that would favor the visualization of the impedance results. For example, in the case of PILs 01 and 03, the points were considerably close; therefore, for better visualization, these dimensions were chosen.
Comments 7: The results reported in Table 3 don’t seem to agree with the plot in Figure 5. For instance, the Rp for PILs01 and PILs02 1000ppm seems lower than 500ppm.
Response 7: Thank you for pointing this information out.
Comments 8: You must also report the X in Table 3
Response 8: Thank you for pointing this information out. We agree with this comment. However, due to the size of Table 3, it was decided to use only the items displayed in it to facilitate the reader's understanding of the data.
Comments 9: Line 434,” Specifically, to calculate inhibition efficiency (I.E) and double-layer capacitance (Cdl), Eqs. (4) – (5) were employed, respectively, where Rp and Rp(inh) represent the polarization resistances in the absence and presence of inhibitors”. Eqs. (8)–(9)
Response 9: Agree. We have, accordingly, changed for Eqs. (9) and (10) (Page 14/32; Line 443). We also corrected the sequence of incorrect equations. (Page 14/32; Lines 452 and 454).
Comments 10: Line 470, “Moreover, the EIS results show that the Cdl values decrease with increasing Rs”. Rp no Rs. Besides, based on question 5, this sentence must be rewritten.
Response 10: Thank you for pointing this information out. We agree with this comment. So we rewrite the sentence like this; The EIS results indicate that increasing Rs reduces Cdl, attributed to the substitution of water and aggressive ions (high ε) by adsorbed PIL molecules (low ε). (Page 16/32; Lines 499 and 501).
Comments 11: Why did you specify the dielectric constant (ε) and the protective film thickness (T), with the letters? You didn’t report Helmholtz's equation.
Response 11: Thank you for highlighting this point. We acknowledge and agree with your comment. In this article, we used the symbols ε (dielectric constant) and T (film thickness) to ensure clarity and consistency when discussing key parameters, aiming to facilitate understanding for the reader. While Helmholtz's equation forms the basis of the relationship between capacitance and these variables, it was not explicitly included, as our focus was on experimental trends rather than theoretical derivations. Nonetheless, we are willing to incorporate it to enhance the theoretical context if deemed necessary.
Comments 12: The optical microscopy images reported in Figure 8 must be changed. You can’t distinguish anything with these images.
Response 12: We agree with this comment. Thank you again for your observation regarding the optical microscopy images in Figure 8. We acknowledge the limitations in image quality, which were due to issues with the optical microscope lenses at the time of image acquisition (during the period of the COVID-19 pandemic). Despite this limitation, the images provided were the best available at the time and, when analyzed alongside the atomic force microscopy (AFM) results, clearly demonstrate the significant differences caused by the addition of corrosion inhibitors on the metal surface. These complementary analyses reinforce the validity of our findings. We appreciate your feedback and understand the importance of high-quality optical microscopy images for clarity and reproducibility in MDPI papers. For future publications, we will ensure improved imaging quality, as we have recently acquired a new optical microscope with enhanced capabilities. This will allow us to provide more detailed and visually clear images in subsequent studies and publications on MDPI.
Comments 13: Line 596, “In contrast, it is observed that a significant reduction in surface damage was observed, with a more uniform appearance and less evidence of corrosive damage”. From the look of the SEM images, it seems that in the presence of the inhibitor, the surface looks etched (Figures B and D); you can see the structure of the metal.
Response 13: Thank you for pointing this information out. We agree with this comment.
In Figure 9 our objective in this case was to emphasize that the addition of protic ionic liquids demonstrated the adsorption of these chemicals (corrosion inhibitors) on the metal surface. This result/phenomenon underpins the metal protection process, which occurs both immediately and progressively over time. To substantiate this point, the surface of the electrode was specifically investigated after the polarization test, where the material's surface undergoes greater wear. This approach allowed us to reinforce our perspective on the protective effects of the protic ionic liquids under these conditions.
Comments 14: Line 470, “Moreover, the EIS results show that the Cdl values decrease with increasing Rs”. Rp no Rs. Besides, based on question 5, this sentence must be rewritten.
Response 14: Consent. We have, accordingly, added the information in Figure 10; (Page 17/32; Line 655). Thank you for your observation regarding the condition of the material's surface after polishing. We hypothesize that some material residue may have remained on the sample after the polishing process, which could have influenced the AFM measurement. Nonetheless, the average surface roughness (ASR) remained within the expected range. According to the literature, polished materials should exhibit a roughness below 20 nm, and our obtained value met this criterion. Therefore, we considered it appropriate to use this result for this study.
In addition, we acknowledge that presenting a sample entirely free of impurities would be ideal. However, the equipment used for sample preparation was not available in our laboratory, and we worked within these constraints to generate the presented results. For future publications, we will strive to obtain a polished sample devoid of impurities to enhance the quality of our analyses. It is important to note that the primary objective of this image was to compare samples with and without the inhibitor, with the polished sample serving only as a control.
Comments 15: Line 647, “As shown in Figure 10, there is An apparent increase in both the roughness and thickness of the carbon steel surface after 24 hours of EXPOSION in saline electrolyte”. How can you see the thickness from these measurements? Plus, correct the English in this sentence.
Response 15: Thank you for pointing this information out. We appreciate you for bringing attention to the issue related to image 10. Due to the error encountered, the average roughness values were initially unavailable, which may have impacted the clarity of the discussion. However, this issue has since been resolved, and the discussion presented in the revised manuscript is now based on the corrected values obtained. These values were analyzed and interpreted using the Gwyddion software to ensure accuracy and reliability in our findings. We appreciate your feedback, as it allowed us to address and clarify this matter, improving the overall quality and comprehensibility of the discussion; As shown in Figure 10, there is an apparent increase in both the roughness and thickness of the carbon steel surface after 24 hours in saline electrolyte. This information indicates that there is a clear and thin protective layer on the metal surface, likely attributed to a physisorption process attributed to the value of free energy of Gibbs (Page 21/32; Line – 677 to 681).
Comments 16: 16. The XRD part is confusing and needs to be rewritten. The author should focus more on the obtained results and explain why they have this type of oxide without mentioning things that are not even related to this study (e.g., line 779, “Still on the modifications of the corrosion products formed on the metal surface, typically occurs under controlled conditions, such as specific temperature ranges (27 to 70°C) and pH values (7.5 to 9.0), because the modification of oxides at alkaline pH and high temperatures occurs due to the higher concentration of OH- ions and the increase in thermal energy, which accelerates oxidation).
Response 16: We appreciate the reviewer’s valuable comments and agree with the observations. To enhance the clarity of the X-ray diffraction section, we will revise certain points to facilitate the reader's understanding of the material. Regarding the specific mention in line 779, we included this discussion because we observed a variation in pH values when PILs were added to the saline solution before testing, particularly with the most effective inhibitor. This observation led us to reference literature addressing the relationship between pH and its potential influence on the formation of the specific phase (goethite). We will ensure this explanation is clearly articulated in the revised manuscript to address the reviewer’s concerns. These points are connected with what is raised in the following paragraph: (Pages 24/32; Lines – 800 to 809). I would like to reiterate, that we truly appreciate the opportunity to contribute to the discussion on corrosion inhibitors and the formation of oxides/oxyhydroxides, which is an ever-scarcer topic in the current literature despite its relevance in studies of atmospheric corrosion. Then, we believe that addressing these aspects enriches the broader understanding of corrosion mechanisms and inhibitor performance. While we acknowledge that certain aspects of our discussion may not be entirely comprehensive, we made every effort to present a thorough and coherent analysis to justify our results and ensure clarity for the reader. We remain open to constructive feedback to further refine and improve our interpretations in this critical area of study.
Comments 17: Line 729, “Interestingly, the phases identified in the blank sample also appeared in the spectrum of PIL 01 (Fig. 12), although with a notable difference: the presence of the goethite phase in the spectrum of PIL 01” the peaks related to goethite are also present in the black spectrum.
Response 17: Thank you for your insightful comment regarding the presence of goethite only in the PIL 01 sample and not in the Blank. This result is consistent with our peak evaluation, where goethite was identified in the PIL 01 sample. Although there is a coinciding peak, this finding was corroborated using Mössbauer and Raman spectroscopy.
We are currently finalizing a microscopic evaluation and employing additional techniques to further investigate the oxides formed on the steel surface. We would like to emphasize that, although goethite might be present in trace amounts on the steel surface in the absence of the inhibitor, this could potentially arise from the natural oxidation of the powder during exposure to the external environment. This is in contrast to the distinct and well-defined peaks observed in PIL 01, which are more clearly spaced when compared to those in PIL 02 and PIL 03. We believe this differentiation underscores the influence of corrosion inhibitors on oxide formation. Finally, these results were not all presented in this article because our objective is to finalize a single article with the evaluation and investigation of the formation of oxides/oxyhydroxides on the surface of the material exposed to saline medium with the addition of protic ionic liquids.
Comments 18: Line 754, “Fuente (2011) indicates that magnetite normally forms a corrosion layer over the surface offering the tiniest protection on steel surfaces [89]. Still, countless researchers have emphasized the protective properties of goethite and ferrihydrite concerning metallic surfaces; however, in this paper experiments involving Protic Ionic Liquids (PILs) revealed only the presence of goethite …”. REVEALED ONLY THE PRESENCE OF GOETHITE, this sentence is confusing, you have all types of oxides.
Response 18: Thank you for your valuable comment regarding the discussion on goethite. While we acknowledge the presence of other oxides/oxyhydroxides in the system (electrolyte/metal/inhibitor), our focus is on explaining the reason behind the formation of this particular oxyhydroxide (goethite) is predominantly observed in systems with the most effective inhibitors. This formation is primarily attributed to the specific adsorption mechanism occurring on the metal surface in a saline medium, as well as the pH around 7.8 to 8.0, which plays a critical role in this process. In systems with PIL 02 and PIL 03, the differing pH (around 5.5 to 6.0) values lead to variations in the oxide/oxyhydroxide phases, resulting in the formation of Magnetite and Lepidocrocite on the metal surface. These oxides/oxyhydroxides are commonly attributed to the corrosion process when carbon steel is exposed to a saline environment. As noted earlier, these oxides/oxyhydroxides are further examined in greater detail in another manuscript I am currently writing. This upcoming article includes characterization techniques such as XRD, Raman spectroscopy, and Mössbauer analysis, along with visual assessments through optical, scanning, and focused ion beam microscopy. One of the key findings in this study is that these oxide layers exhibit varying thicknesses and porosities, and have an impact on the permeation of elements such as Chlorine (Cl-). Given this, I believe the current discussion sufficiently prepares readers for the more detailed findings that will be published in the future. Therefore, I am confident that the material presented holds academic relevance and warrants publication in the journal to which it has been submitted.
4. Response to Comments on the Quality of English Language
Point 1:
We have no further considerations
5. Additional clarifications
We have no further considerations

Reviewer 3 Report (New Reviewer)
Comments and Suggestions for Authors
In the present work, the authors studied the use of protic ionic liquids as corrosion inhibitor for carbon steel, the electrochemical characterization shows good inhibition performances and the authors discuss extensively the corrosion inhibition mechanist.
However, the following should be addressed to improve the manuscript.
1. In the introduction the authors wrote: “This particular material has resistance to procedures associated with high temperature and pressure; however, it has low chemical resistance against corrosion due to the steel composition” it is no clear if this phrase refers to stainless or carbon steel, it must be specified.
2. Also, in the introduction the authors wrote: “has been studied little in relation to its corrosion behavior in aggressive environments such as the saline environment investigated in this article.” this is not true, there is a vast literature on the corrosion mechanisms of carbon steels, including ASTM A36 steel.
3. In chapter “2.3. Sample preparation for general evaluations” the authors wrote: “Initially, the 148 steel samples (A36) with dimensions of one cm2” change for: “Initially, the 148 steel samples (A36) with dimensions of 1 cm2” use numeric characters.
4. Axes of Nyquist plots must be equal-sized for the same sample, for better visualization of the loops.
Comments on the Quality of English LanguageI suggest revising the English of the text
Author Response
1. Summary
Thank you for your insightful and positive feedback on our manuscript regarding corrosion inhibitors (PILs). We sincerely appreciate the time and effort you devoted to reviewing our manuscript and recognizing its contributions to a better final work. In this regard, in response to your comments, we have carefully revised the manuscript, incorporating the necessary corrections, which are marked in the re-submitted files using track changes. Additionally, we apologize for any grammatical errors, oversights in the equations, or low-quality images in the original submission.
3. Point-by-point response to Comments and Suggestions for Authors
Comments 1: In the introduction the authors wrote: “This particular material has resistance to procedures associated with high temperature and pressure; however, it has low chemical resistance against corrosion due to the steel composition” it is not clear if this phrase refers to stainless or carbon steel, it must be specified.
Response 1: Thank you for pointing this information out. We agree with this comment. Therefore, we have changed the unity according to the review for carbon steel (ASTM A36), (Page – 2/32; Line – 48).
Comments 2: Also, in the introduction the authors wrote: “has been studied little in relation to its corrosion behavior in aggressive environments such as the saline environment investigated in this article.” this is not true, there is a vast literature on the corrosion mechanisms of carbon steels, including ASTM A36 steel.
Response 2: Thank you for pointing this information out. We agree with this comment. We add this information, ASTM A36 steel is extensively utilized in industry due to its structural properties, in particular excellent mechanical strength, and broad market availability. While its corrosion behavior has been widely investigated, its interaction with specific sustainable inhibitors, such as protic ionic liquids, in aggressive environments like saline conditions remains insufficiently explored in the literature. (Page – 2/32; Line – 89 to 93).
Comments 3: In the chapter “2.3. Sample preparation for general evaluations” the authors wrote: “Initially, the 148 steel samples (A36) with dimensions of one cm” change to: “Initially, the 148 steel samples (A36) with dimensions of 1 cm” use numeric characters.
Response 3: Agree. We have, accordingly, changed this sentence: Initially, the steel samples (A36) with dimensions of one cm2 were sanded with emery grit paper 120, 220, 400, 600, and 1.200, without further polishing (Page – 4/32; Line – 150 to 152).
Comments 4: " Axes of Nyquist plots must be equal sized for the same sample, for better visualization of the loops´´.
Response 4: Thank you for your thorough evaluation of our manuscript and your valuable comments. We appreciate your feedback; The modification has been made. (Page – 13/32; Line – 434 to 435).
4. Response to Comments on the Quality of English Language
Point 1:
We have no further considerations
5. Additional clarifications
We have no further considerations

Round 2
Reviewer 2 Report (Previous Reviewer 3)
Comments and Suggestions for Authors
The authors answered all questions satisfactorily. The manuscript may be published in its current form.
This manuscript is a resubmission of an earlier submission. The following is a list of the peer review reports and author responses from that submission.
Round 1
Reviewer 1 Report
Comments and Suggestions for Authors
The work lacks innovation, with minimal contribution to advancing the field. The experimental data are rough and inconsistently presented, and the electrochemical curves appear disorganized, making it difficult to interpret the results effectively. Additionally, the fitting is inaccurate and lacks proper validation, raising concerns about the reliability of the conclusions. Overall, the manuscript falls far short of the standards required for an SCI paper in terms of both scientific rigor and clarity of presentation.
Comments on the Quality of English Languagelow
Reviewer 2 Report
Comments and Suggestions for Authors
In this paper, the authors studied three distinct ethanolamine-based protic ionic liquids as corrosion inhibitors for carbon steel in saline environments.
The topic is interesting and I consider it important because of the need for corrosion inhibitors in industry. Protic ionic liquids seem to be an alternative to consider, so the article may be of interest to the scientific community.
The manuscript is organized and presented in a well-structured manner. The text is written in a way that the reader can understand.
The research methodology is appropriate. The graphs, tables, and images correctly present the result obtained. The data are interpreted appropriately and consistently. The results are properly discussed and explained to the reader.
The literature contains 101 items. There are no inappropriate self-citations by the authors.
However, the manuscript contains many errors and needs improvement. My detailed comments, in the order in which the shortcomings appear in the manuscript:
1. line 146
It is generally accepted that the unit of corrosion rate is written mm/y, not mmy.
2. The manuscript lacks an explanation of what the symbol k in equation (2) means.
3. The designation of the inhibition efficiency in line 151 (% IEWL) is inconsistent with the designation in equation (3) - i.e. %IE.
4. line 155
"Where ʋcorr(b) and ʋcorr(i) are the corrosion rates with inhibitors and without them."
The description should be the reverse: ʋcorr(b) is the corrosion rate without inhibitor, not with inhibitor.
5. line 161
For electrochemical measurements, the authors used a potentiostat rather than a galvanostat.
6. line 162
What is missing is that the ppm values given in this sentence are for PILs.
7. line 165
The text of the manuscript lacks information on the amplitude of the AC perturbation signal in EIS measurements.
8. The title of section 3.1 should say PILs, not ILs.
9. The designations given in the rows of Table 1 are different from those given in line 235. E.g. in line 235 is SD, and in the table is ΔW.
10. Figure 4
The X axes titles should say volts, not millivolts.
11. line 349
"that extend over the total initial cathodic stages (-0.2 mV) to the anodic region (2 mV)"
Incorrectly written mV, instead of V.
It should be: "that extend over the total initial cathodic stages (-0.2 V) to the anodic region (0.2 V)"
12. line 354
The text of the manuscript lacks information on how the corrosion current values were determined.
13. line 398
To avoid ambiguity, efficiency should be designated by a different symbol than the overvoltage η.
14. The manuscript lacks information on how the efficiency values given in Table 2 were determined.
15. line 400
Incorrect statement “efficiency through the polarization technique”.
Efficiency was achieved through the use of the inhibitor, not through the polarization technique.
16. line 407
"Impedance Spectroscopy (EIS)"
It should be: Electrochemical Impedance Spectroscopy (EIS)
17. line 446
The numbering of the equations should be corrected. The equation in line 446 has the same number (8) as the equation in line 289. The subsequent equations should be renumbered.
18. equation in line 446
The given formula for calculating efficiency is wrong.
It should be:
(Rp - Rp.inh) / Rp
19. title of Figure 8
"ppm of each inhibitor PIL 01 (2-HEAF), PIL 03 (2-HEAP), and PIL 05 (2-HEAPe)"
It should be: PIL 01, PIL 02, PIL 03 - not PIL 01, PIL 03, PIL 05.
20. Figure 10
Descriptions A, B and C above the 3D images are invisible.
21. line 845
The equation number is wrong.
Reviewer 3 Report
Comments and Suggestions for Authors
Although one of the inhibitors showed good inhibition efficiency, the manuscript is just a routine study about using protic ionic liquids. The effect of temperature and long periods of exposure to the saline solution was not even taken into consideration. The effectiveness should be significantly reduced since the tested inhibitors show physical adsorption at high temperatures. The authors failed to describe the novelty and state-of-the-art about the protic ionic liquids.
1. English needs to be amended. There are some grammatical and typos in the text.
2. The introduction needs to be improved. The authors must state the novelty and state-of-the-art about the protic ionic liquids.
3. “K” in equation 2 is not specified.
4. Check line 155, “Where Ê‹corr(b) and Ê‹corr(i) are the corrosion rates with inhibitors and without them”. In this way, the efficiency will be negative.
5. Correct line 235, “A - Average / W - Weight (1) and (2) / Ê‹corr - Corrosion Rate / SD - Standard Deviation”. This sentence in the caption is not necessary
6. The X and Y axes in the Nyquist plot must be the same. You must also report Bode plots, together with the Nyquist, to prove how many time constants you have.
7. The results reported in Table 3 don’t seem to agree with the plot in Figure 5. For instance, the Rp for PILs01 and PILs02 1000ppm seems lower than 500ppm.
8. You must also report the X2 in Table 3.
9. Line 434, ”Specifically, to calculate inhibition efficiency (I.E) and double-layer capacitance (Cdl), Eqs. (4)–(5) were employed, respectively, where Rp and Rp(inh) represent the polarization resistances in the absence and presence of inhibitors”. Eqs. (8)–(9)
10. Line 470, “Moreover, the EIS results show that the Cdl values decrease with increasing Rs”. Rp no Rs. Besides, based on question 5, this sentence must be rewritten.
11. Why did you specify the dielectric constant (ε) and the protective film thickness (T), with the letters? You didn’t report Helmholtz's equation.
12. The optical microscopy images reported in Figure 8 must be changed. You can’t distinguish anything with these images.
13. Line 596, “In contrast, it is observed that a significant reduction in surface damage was observed, with a more uniform appearance and less evidence of corrosive damage”. From the look of the SEM images, it seems that in the presence of the inhibitor, the surface looks etched (Figures B and D); you can see the structure of the metal.
14. The AFM images need to be improved. They lack a proper scale, and they do not seem to be well prepared. Something is present on the polished sample surface, likely some unremoved polished materials
15. Line 647, “As shown in Figure 10, there is A apparent increase in both the roughness and thickness of the carbon steel surface after 24 hours of EXPOSION in saline electrolyte”. How can you see the thickness from these measurements? Plus, correct the English in this sentence.
16. The XRD part is confusing and needs to be rewritten. The author should focus more on the obtained results and explain why they have this type of oxide without mentioning things that are not even related to this study (e.g., line 779, “Still on the modifications of the corrosion products formed on the metal surface, typically occurs under controlled conditions, such as specific temperature ranges (27 to 70°C) and pH values (7.5 to 9.0), because the modification of oxides at alkaline pH and high temperatures occurs due to the higher concentration of OH- ions and the increase in thermal energy, which accelerates oxidation).
17. Line 729, “Interestingly, the phases identified in the blank sample also appeared in the spectrum of PIL 01 (Fig. 12), although with a notable difference: the presence of the goethite phase in the spectrum of PIL 01” the peaks related to goethite are also present in the black spectrum.
18. Line 754, “Fuente (2011) indicates that magnetite normally forms a corrosion layer over the surface offering tiniest protection on steel surfaces [89]. Still, countless researchers have emphasized the protective properties of goethite and ferrihydrite concerning metallic surfaces; however, in this paper experiments involving Protic Ionic Liquids (PILs) revealed only the presence of goethite …”. REVEALED ONLY THE PRESENCE OF GOETHITE, this sentence is confusing, you have all types of oxides.
Comments on the Quality of English Language
English needs to be amended. There are some grammatical and typos in the text.